# Protein-induced membrane strain drives supercomplex formation

**Maximilian C Pöverlein[†], Alexander Jussupow[†], Hyunho Kim, Ville RI Kaila\***

Department of Biochemistry and Biophysics, Stockholm University, Stockholm, Sweden

## eLife Assessment

In this **important** study, the authors conducted extensive atomistic and coarse-grained simulations as well as a lattice Monte Carlo analysis to probe the driving force and functional impact of supercomplex formation in the inner mitochondrial membrane. The study highlighted the major contribution from membrane mechanics to the supercomplex formation and revealed interesting differences in structural and dynamical features of the protein components upon complex formation. Upon revision, the analysis is considered **solid**, although the magnitude of estimated membrane deformation energies seem somewhat large. Overall, the study is thorough, creative and the impact on the field of bioenergetics is expected to be significant.

**\*For correspondence:**
ville.kaila@dbb.su.se

[†]These authors contributed equally to this work

**Competing interest:** The authors declare that no competing interests exist.

**Abstract** Mitochondrial membranes harbor the electron transport chain (ETC) that powers oxidative phosphorylation (OXPHOS) and drives the synthesis of ATP. Yet, under physiological conditions, the OXPHOS proteins operate as higher-order supercomplex (SC) assemblies, although their functional role remains poorly understood and much debated. By combining large-scale atomistic and coarse-grained molecular simulations with analysis of cryo-electron microscopic data and statistical as well as kinetic models, we show here that the formation of the mammalian $I/III_2$ supercomplex reduces the molecular strain of inner mitochondrial membranes by altering the local membrane thickness and leading to an accumulation of both cardiolipin and quinone around specific regions of the SC. We find that the SC assembly also affects the global motion of the individual ETC proteins with possible functional consequences. On a general level, our findings suggest that molecular crowding and strain effects provide a thermodynamic driving force for the SC formation, with a possible flux enhancement in crowded biological membranes under constrained respiratory conditions.

## Introduction

Biological electron transport chains (ETC) comprise a series of membrane-bound enzyme complexes (CI-CIV) that transfer electrons toward oxygen and protons across a biological membrane, creating a proton motive force (PMF) that powers the synthesis of ATP and active transport (***Mitchell, 1961***; ***Kaila and Wikström, 2021***). Biological membranes are often envisaged in the light of the classical fluid mosaic model (***Singer, 1972***), where the membrane proteins and lipids independently diffuse as a two-dimensional solution. However, the biological membranes are highly crowded (***Kirchhoff, 2008***; ***Capaldi, 1982***; ***Hatefi, 1985***), particularly the inner mitochondrial membrane (IMM), with a protein content of around 45% (***Schlame, 2021***). In this regard, the majority of the respiratory complexes do not diffuse independently within the membrane but operate rather as larger supercomplex (SC) assemblies, as first revealed by blue native gels (***Schägger and Pfeiffer, 2000***) and confirmed by structural analyses (***Schäfer et al., 2007***; ***Althoff et al., 2011***; ***Dudkina et al., 2011***). Moreover,

several lipid molecules, particularly cardiolipin, a central anionic lipid of the IMM, are often tightly bound to the SCs (*Berndtsson et al., 2020*; *Pfeiffer et al., 2003*; *Hryc et al., 2023*; *Riepl et al., 2024*). Yet, despite these structural insights, the functional role of the SC assemblies and the physical principles leading to their formation remain elusive and much debated.

SCs have been suggested to provide kinetic advantages (*Berndtsson et al., 2020*; *Moe et al., 2021*), favor substrate channeling (*Althoff et al., 2011*; *Lenaz et al., 2016*, but cf. *Blaza et al., 2014*), decrease the formation of reactive oxygen species (ROS) (*Diaz et al., 2012*; *Maranzana et al., 2013*), stabilize individual proteins (*Schägger et al., 2004*; *Acín-Pérez et al., 2004*; *Letts and Sazanov, 2017*; *Protasoni et al., 2020*), and/or reduce non-specific protein–protein interactions (*Blaza et al., 2014*). It has also been recognized that the high protein density in membranes may play an important role in the SC formation (*Blaza et al., 2014*), and influence, for example, the lipid curvature (*Schlame, 2021*; *Brown, 2017*). On a physiological level, decreased SC formation has been linked to various pathophysiological conditions such as diabetes (*Antoun et al., 2015*), heart failure (*Rosca et al., 2008*), and apoptosis (*Cogliati et al., 2013*), although recent experiments (*Milenkovic et al., 2023*) on mice unable to form SC showed no significant differences relative to WT mice under the studied conditions.

In recent years, cryo-electron microscopic (cryo-EM) and cryo-electron tomographic studies revealed the structure of several SCs, including the mammalian 1.5 MDa SCI/III$_2$ and the 1.7 MDa respirasome (SCI/III$_2$/IV) (*Gu et al., 2016*; *Wu et al., 2016*; *Letts et al., 2016*; *Guo et al., 2017*; *Davies et al., 2018*), as well as various exotic megacomplexes isolated from single-celled eukaryotes (*Zhou et al., 2022*; *Mühleip et al., 2023*). SCs are also essential for some bacteria, such as the SCIII$_2$/IV$_2$ of actinobacteria (*Riepl et al., 2024*; *Wiseman et al., 2018*; *Gong et al., 2018*; *Kao et al., 2016*) that catalyze quinol oxidation coupled to oxygen reduction in a tight obligate protein assembly. Inhibition of such bacterial SCs provides potential avenues for treating pathogenic infections, such as tuberculosis, with emerging multi-drug-resistant strains.

To probe the physical principles leading to the SC formation and its molecular consequences, we study here the structure and dynamics of the mammalian mitochondrial SCI/III$_2$ by combining large-scale atomistic and coarse-grained molecular dynamics (cgMD) simulations with analysis of cryo-EM

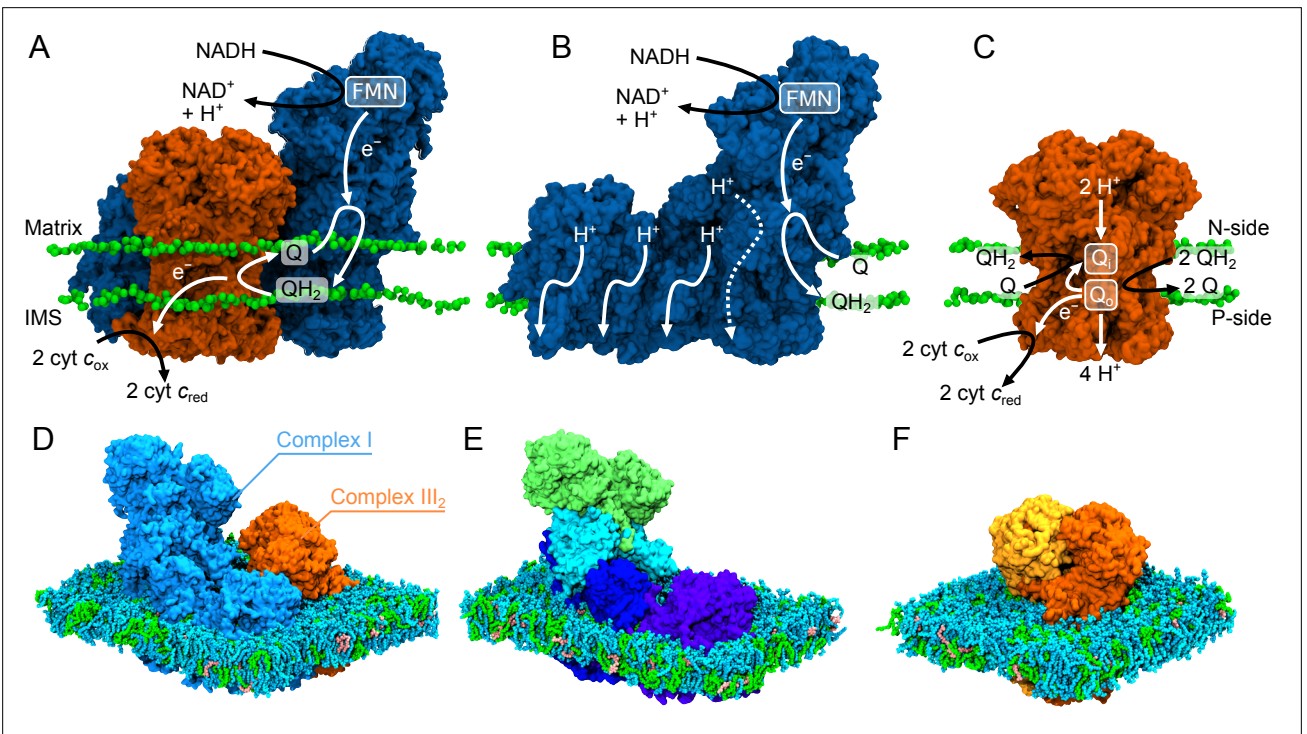

**Figure 1.** Structure and function of Complex I, Complex III$_2$, and the SCI/III$_2$. (**A**) The structure of the OXPHOS complexes with lipid headgroups from atomistic molecular dynamics (MD) simulations. (**B, C**) Overview of the structure and function of redox-driven proton pumping in CI and the Q-cycle of CIII$_2$. Molecular models of the (**D**) SCI/III$_2$, (**E**) CI, and (**F**) CIII$_2$ used in the atomistic MD simulations (water and ions were omitted for visual clarity).

data and mathematical models that provide insight into how crowding effects, membrane strain, deformation effects, as well as dynamic protein–lipid interactions influence SC formation.

The studied SCI/III$_2$ catalyzes an NADH-driven quinone/cytochrome $c$ reduction and offers a unique system to probe how the quinone (Q) substrate is transported within the SC assembly, and how the individual proteins interact within the membrane. In this regard, the quinol (QH$_2$) produced by the CI module of the SC is re-oxidized by the CIII$_2$ module, while the proton transport activity (CI: 2H$^+$/e$^-$; CIII$_2$: 1H$^+$/e$^-$) powers the synthesis of ATP (*Figure 1*). Interestingly, prior cryo-EM studies of this SC (*Letts et al., 2019*) revealed an asymmetric protein assembly with a trapped quinone in the proximal/distal Q$_o$ site, suggesting possible functional consequences. However, the dynamics and interaction of the SC and its substrates within the surrounding membrane environment and the individual active site remain poorly understood.

## Results
### Protein–membrane interactions modulate the lipid dynamics and membrane strain

To probe how the SC formation affects the dynamics of the OXPHOS proteins, the lipid membrane, and the quinone pool, we studied the ovine SCI/III$_2$ by both atomistic molecular dynamics (aMD) and cgMD simulations. While the aMD simulations give insight into the microsecond dynamics (11 µs in total, *Appendix 1—table 2*) of the system with atomic details, our cgMD simulations allowed us to probe the SC dynamics and its interactions in the membrane on much longer timescales (~0.3 ms in cumulative simulation time, 50–75 µs/simulation) and with larger membrane size at a more approximate level (see Methods). The SC models were further compared against simulations of the individual CI and CIII$_2$ complexes, as well as simulations of a lipid membrane, with a POPC:POPE:CDL ratio (2:2:1) as in the IMM. In the following, we define *membrane strain* as the local perturbations of the lipid bilayer induced by protein–membrane interactions. These include changes in (1) membrane thickness, (2) the local membrane composition, (3) lipid chain configurations, and (4) local curvature of the membrane plane relative to an undisturbed, protein-free bilayer. Together, these phenomena reflect the thermodynamic effects associated with accommodating large protein complexes within the membrane.

We find that the individual OXPHOS complexes, CI and CIII$_2$, induce pronounced membrane strain effects, supported both by our aMD (*Appendix 1—figure 2A*) and cgMD simulations with a large surrounding membrane (*Figure 2G*). These effects persist for both individual complexes and assembled SC. We observe a local decrease in the membrane thickness at the protein–lipid interface (*Figure 2G*, *Appendix 1—figure 2A, D, E*), likely arising from the thinner hydrophobic belt region of the OXPHOS proteins (ca. 30 Å, *Appendix 1—figure 1A*) relative to the lipid membrane (40.5 Å, *Appendix 1—figure 1D*). We further observe ~30% accumulation of cardiolipin at the thinner hydrophobic belt regions (*Figure 2H*, *Appendix 1—figure 2B, F, G*), with an inhomogeneous distribution around the OXPHOS complexes. While specific interactions between CDL and protein residues may contribute to this enrichment (*Figure 2N*), CDL also thermodynamically prefers thinner membranes (~38 Å, *Appendix 1—figures 1D and 5E*). These changes are further reflected in the reduced *end-to-end* distance of lipid chains in the local membrane belt (see Methods, *Appendix 1—figure 6*, cf. also *Venable et al., 2015*; *Chadda et al., 2021*; *Mondal et al., 2013*; *Lundbaek et al., 2010*). In addition to the perturbations in the local membrane thickness, the OXPHOS proteins also induce a subtle inward curvature toward the protein–lipid interface (*Appendix 1—figure 5G*), which could modulate the accessibility of the Q/QH$_2$ substrate into the active sites of CI and CIII$_2$ (see below, section *Discussion*). This curvature is accompanied by a distortion of the local membrane plane itself (*Figure 2A–F*, *Appendix 1—figures 4A–C and 8*), with perpendicular leaflet displacements reaching up to ~2 nm relative to the average leaflet plane.

To quantify the membrane strain effects, we analyzed the cgMD trajectories by projecting the membrane surface onto a two-dimensional grid and calculating the local membrane height and thickness at each grid point. From these values, we quantified the *local membrane curvature* (*Appendix 1—figure 5H*), which measures the energetic cost of bending the membrane from a flat geometry ($\Delta G_{curv}$) based on the Helfrich model (*Helfrich, 1973*; *Campelo et al., 2014*). We also computed the energetics associated with changes in the *membrane thickness,* assessed from the deviations from an ideal

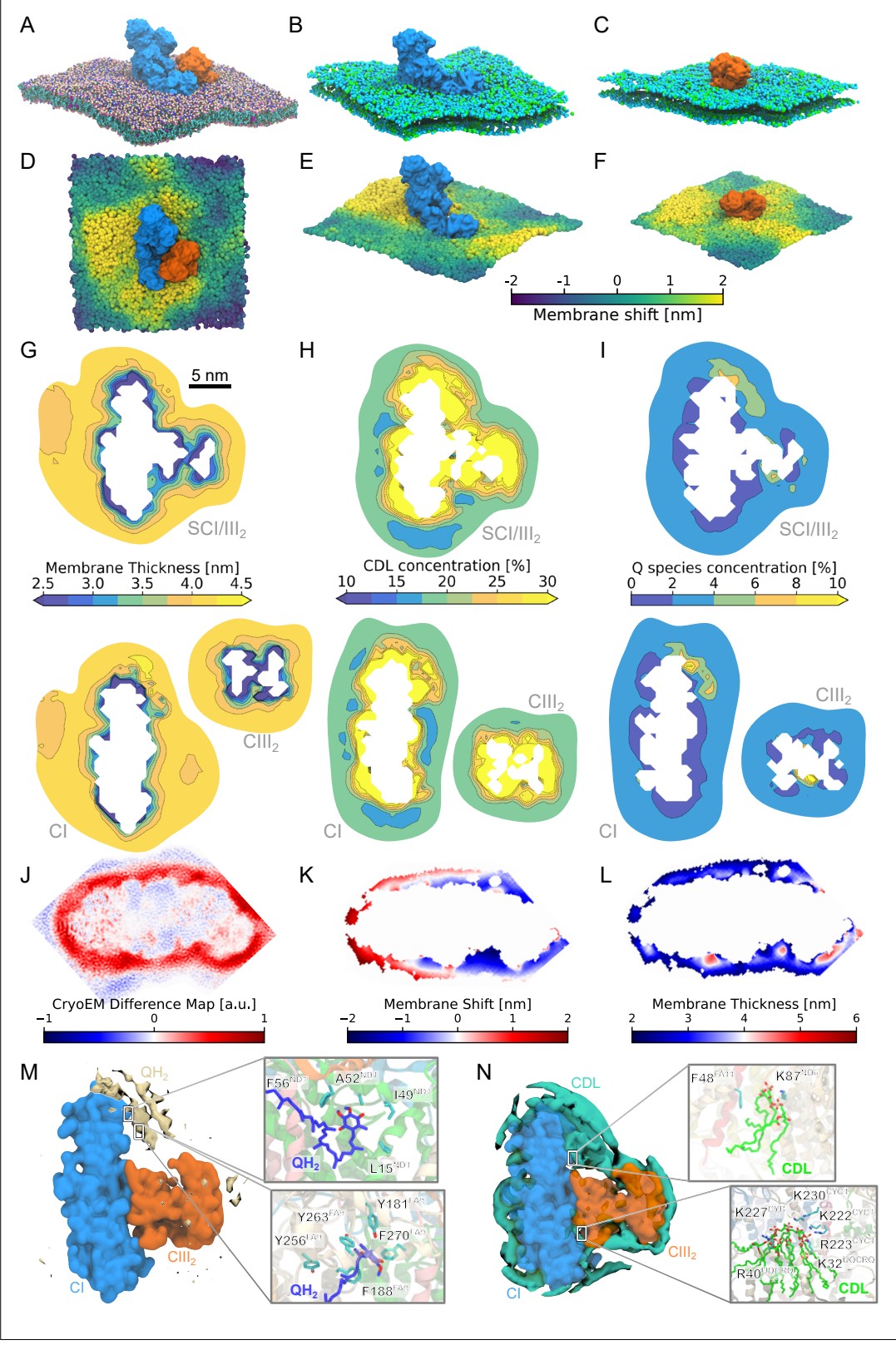

**Figure 2.** SC formation affects the concentration of lipids and the membrane thickness. (**A–C**) OXPHOS complexes viewed from the N-side of the membrane. Membrane shift induced by (**D**) the SC, and (**E, F**) the isolated CI and CIII$_2$. The membrane shift relative to the average membrane plane is colored by the shift in z-position viewed from the N-side. (**G**) Local membrane thickness around SC (top), CI and CIII$_2$ (bottom). (**H**) Local cardiolipin

*Figure 2 continued on next page*

*Figure 2 continued*

concentration. (**I**) Local concentration of Q (Q + QH$_2$). (**J**) Cryo-electron microscopic (cryo-EM) difference map. (**K**) Membrane shift determined from the cryo-EM map. (**L**) Membrane thickness determined from cryo-EM map. (**M**) Density of Q/QH$_2$ (headgroup) from coarse-grained molecular dynamics (cgMD) simulations. *Insets*: Protein–Q interactions from atomistic molecular dynamics (MD). (**N**) Density of cardiolipin (headgroup) from cgMD simulations. *Insets*: Protein–CDL interactions from atomistic MD.

local membrane in the absence of embedded proteins ($\Delta G_{thick}$, see Appendix, for technical details). Our analysis suggests that both contributions are substantially reduced upon formation of the SC, with the curvature penalty decreasing by 79.2 ± 5.2 kcal mol$^{-1}$ (for a membrane area of ca. 1000 nm$^2$) and the thickness penalty by 2.8 ± 2.0 kcal mol$^{-1}$ (*Appendix 1—table 1*). These results indicate a significant thermodynamic advantage for SC formation, as it minimizes lipid deformation and stabilizes the membrane environment surrounding Complexes I and III$_2$.

Remarkably, the SC assembly partially releases this strain energy as a result of a smaller solvation area established at the SC–lipid interface (*Figure 2G*, *Appendix 1—figures 5 and 8*, *Appendix 1—table 5*). We find that an increase in the lipid tail length decreases the relative stability of the SC (*Appendix 1—figure 5F*), further supporting that the hydrophobic mismatch between the OXPHOS proteins and the lipid membrane modulates the strain effect (*Appendix 1—figure 5A–C*).

Taken together, the analysis suggests that the OXPHOS complexes affect the mechanical properties of the membranes by inducing a small inwards curvature toward the protein–lipid interface (*Appendix 1—figure 5G, H*), resulting in a membrane deformation effect, while the SC formation releases some deformation energy relative to the isolated OXPHOS complexes. The localization of specific lipids around the membrane proteins, as well as local membrane perturbation effects, is also supported by simulations of other membrane proteins (*Corradi et al., 2018*; *Baratam et al., 2021*), suggesting that the effects could arise from general protein–membrane interactions.

Although the current cryo-EM maps do not resolve the molecular details of the surrounding lipid membrane (but see below), we performed a spatial integration of the experimental cryo-EM density map by training a neural network model (see Methods) that allowed us to identify and calculate the membrane thickness of the lipid belt around Complex I (PDB ID: 6RFR, EMD-4873) (*Parey et al., 2019*). In this regard, we find that the experimental data also shows a statistically significant membrane thinning around CI relative to the SC at the protein–membrane interface (*Figure 2J–L*), supporting our simulation results. We note that during the finalization of this work, a membrane distortion effect was also reported for in situ high-resolution cryo-EM structures of respiratory SCs (*Zheng et al., 2024*), thus providing additional support for our findings.

We observe that the SC assembly influences the lipid and water dynamics at the protein–lipid interface and perturbs the local dielectric at the membrane plane (*Appendix 1—figure 1E*). The dielectric constant ($\varepsilon_{\perp}$) shows a local increase up to 1.5 nm from the membrane before it decays to the aqueous bulk dielectric (*Appendix 1—figure 1C, E*). We note that this effect is strongly affected by the lipid type, particularly by the CDL that accumulates around the SC (*Figure 2H*, *Appendix 1—figure 2B*) and creates a microenvironment, which could enhance a local proton gradient.

Taken together, our combined findings suggest that the SC formation is affected by thermodynamic effects that reduce the molecular strain in the lipid membrane, while the perturbed microenvironment also affects the lipid and Q dynamics, as well as the dynamics of the OXPHOS proteins (see below).

## The SC assembly alters the quinone dynamics but does not support substrate channeling

In addition to the phospholipids, the IMM contains 3–5% ubiquinone (Q), which functions as the electron carrier between CI and CIII$_2$ in the form of ubiquinol (QH$_2$). Our simulations, performed with ca. 5% Q/QH$_2$ (1:1), suggest that the long isoprenoid tail of ubiquinone (Q$_{10}$) has similar physico-chemical properties as the lipid tails, while the Q/QH$_2$ headgroup is polar and localizes at the membrane surface, with a flip-flop rate between the leaflets of ca. 100–150 ns (*Appendix 1—figure 3A–C*). The fast flip-flop motion could support the quinol diffusion from CI to CIII$_2$, in which the quinol exits the negatively charged (N-side/matrix side) membrane surface of CI and enters the Q$_o$ site of CIII$_2$ near the positively charged (P-side/intermembrane space) side of the membrane (see *Discussion*). Interestingly, the Q pool does not accumulate uniformly around the SC. Instead, we observe a ca. 3%

increase in the local Q concentration near the substrate binding sites of both CI and CIII$_2$ relative to the bulk membrane, while the immediate OXPHOS surroundings show an overall Q/QH$_2$ depletion (*Figure 2I*, *Appendix 1—figures 2C and 7D, E*). This local Q pool could arise from specific interactions between Q and the OXPHOS proteins, for example, in subunits ND1 and NDUFA9 of CI that contain several non-polar residues and surface charges that interact with the quinones (*Figure 2M*, *Appendix 1—figure 14*). In contrast, on the proximal side of CIII$_2$, we observe a subtle decrease in the local Q concentration relative to CIII$_2$ alone (*Figure 2I*, *Appendix 1—figures 2C and 7D, E*), an effect that could arise from a shift in the membrane plane near the Q$_o$ site and affect the substrate access to the proximal Q$_o$ site (*Appendix 1—figures 4 and 8*). These observations are consistent with the occupied distal Q$_o$ site and empty proximal Q$_o$ site observed in the cryo-EM structure of the SCI/III$_2$ (*Letts et al., 2019*).

Overall, while our simulations indicate that quinones accumulate around the OXPHOS proteins, our data do not support substrate channeling (*Althoff et al., 2011*) (see *Discussion*), as we observe neither a contiguous region of an elevated Q pool between the complexes, nor a directed diffusion pathway of the quinol from CI to CIII$_2$ (*Figure 2I*, *Appendix 1—figure 3D–I*). Moreover, the locally hindered access to the proximal Q$_o$ site due to the membrane shift could further hamper the substrate binding into the active site (*Appendix 1—figure 4D–E*).

## The SC assembly alters the conformational dynamics of CI and CIII$_2$

We next probed how the SC assembly affects the dynamics of the individual OXPHOS proteins based on our atomistic MD simulations of CI, CIII$_2$, and the SCI/III$_2$. Despite the large (1.5 MDa) SC, we note that the SC assembly is stabilized by only a few specific hydrogen-bonding and charged interactions at the interface of NDUFB9/NDUFB4 and UQCRC1, particularly the Arg29[FB4]-Asp260[C1]/Glu259[C1] ion-paired network that interacts by ca. –10 kcal mol$^{-1}$ during the MD simulations (*Figure 3A, C*). Indeed, deletion of this Glu258-Asp260 region in UQCRC1 led to a drastic reduction in the stability of SC in mice (*Milenkovic et al., 2023*) (see *Appendix 1—figure 11*), thus supporting the importance of these interactions in the SC.

Based on essential dynamics analysis (see Methods, *Videos 1–5*) that projects out dominant protein motion, we find that the CI and CIII$_2$ modules of the SC undergo a back-and-forth *rocking* motion around the interface region, leading to a *breathing* motion between the hydrophilic domains of the OXPHOS complexes on the N-side of the membrane (matrix side) (*Figure 3B*, Mode 1, *Video 1*), while the second dominant motion (Mode 2, *Video 1*) couples the *opening/closing* motion of CI (cf. also *Jussupow et al., 2019*) with a *rocking* motion of CIII$_2$. The third mode arises from a combination of the *twisting* motion of the hydrophilic domain of CI and a minor *rocking* motion of CIII$_2$ (Mode 3, *Video 1*), which could influence the membrane accessibility of the Q sites. In this regard, our graph theoretical analysis (*Appendix 1—figure 11C, D*) further indicates that ligand binding to Complex I induces a dynamic crosstalk between NDUFA5 and NDUFA10, consistent with previous work (*Grba and Hirst, 2020*; *Di Luca and Kaila, 2018*), and affecting also the motion of UQCRC2 with respect to its surroundings. Taken together, these effects suggest that the dynamics of CI and CIII$_2$ show some correlation that could result in allosteric effects, as also indicated based on cryo-EM analysis (*Letts et al., 2019*).

We note that the SC assembly induces a subtle conformational change in CI that increases the angle between the hydrophilic and membrane domains (*Figure 3F*). This motion is linked to the *active/deactive* transition (A/D) (*Kampjut and Sazanov, 2020*), which regulates CI activity, particularly under hypoxic/anoxic conditions, and hinders reverse electron transfer (*Kampjut and Sazanov, 2020*; *Kotlyar and Vinogradov, 1990*; *Fiedorczuk et al., 2016*; *Blaza et al., 2018*; *Zhu et al., 2016*; *Babot et al., 2014*) (but cf. also *Chung et al., 2022*). These changes in CI lead to a subtle shift in the conformation of key transmembrane helices (TM) and surrounding conserved loops. In the SCI/III$_2$, the TM3 of subunit ND6 undergoes a twist toward the *deactive* state (*Figure 3H*, *Appendix 1—figure 13A–D*), while the β1–β2 loop of subunit NDUFS2 exhibits lower flexibility in both the *apo* and the QH$_2$ bound state relative to CI alone (*Appendix 1—figure 13E–H*). These regions are central in modulating the coupling between the proton transport and the electron transfer activities (*Röpke et al., 2021*, cf. also *Kim et al., 2023*), and the dynamics of these regions could thus have functional consequences. CI also shows a dominant vibrational motion that *twists* the hydrophilic domain relative to the membrane domain (*Figure 3B, D*), and samples a wider angle between hydrophilic and membrane domains in

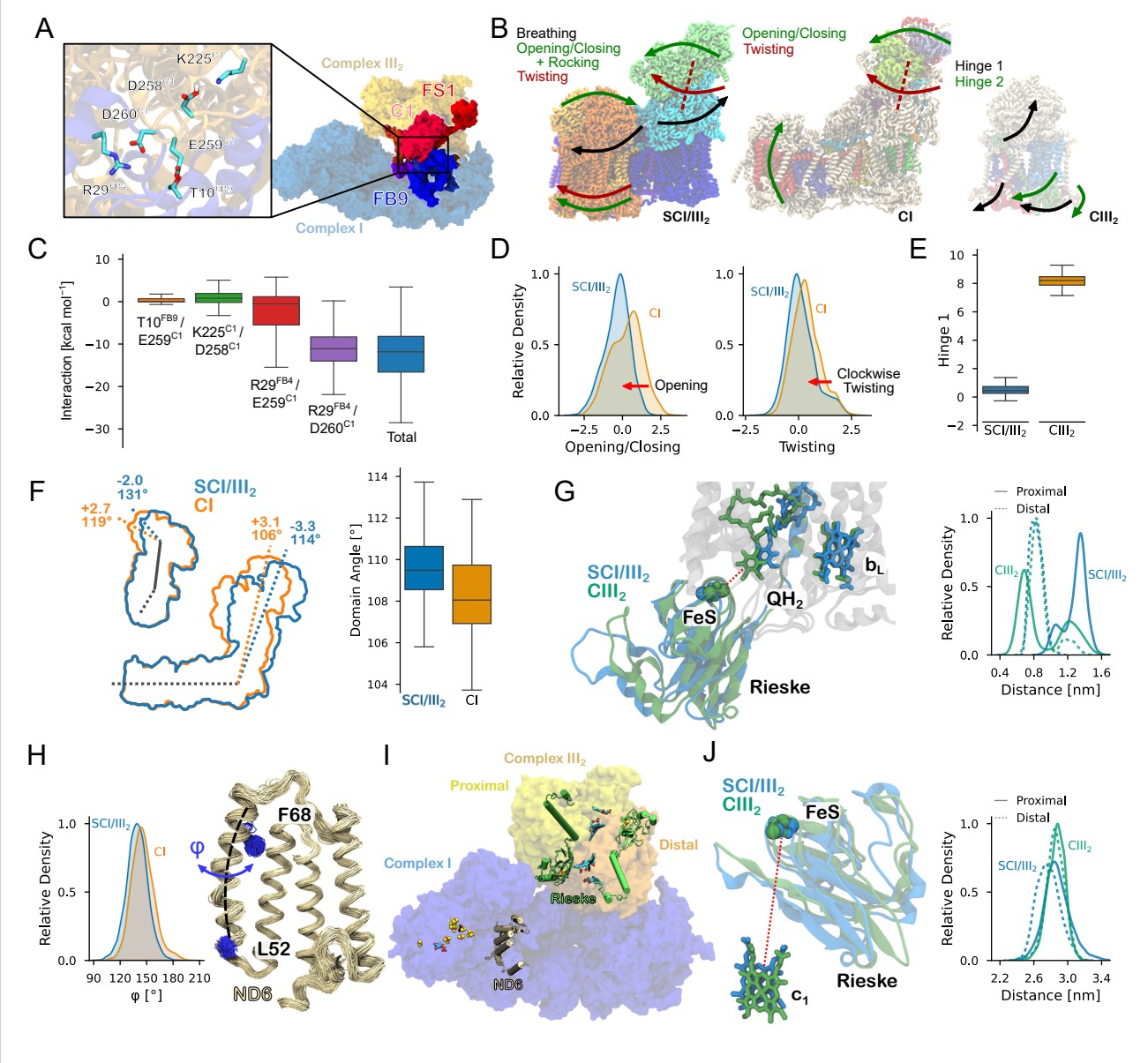

**Figure 3.** Conformational dynamics of the OXPHOS proteins and the SC. (**A**) Subunits comprising the SC interface. *Inset*: Interaction near the DED-loop. (**B**) Normal modes derived from essential dynamics analysis for the SC (*left*), CI (*middle*), and CIII$_2$ (*right*). See *Videos 1–5* for normal modes of the SC, CI, and CIII$_2$. (**C**) Decomposition of interaction energies within the SC. (**D**) Distribution of the *opening/closing* mode (*left*) and the *twisting* mode (*right*) for the isolated CI (in orange) and the CI within the SC (in blue). (**E**) Differences in the dynamics of CIII$_2$ for minimum and maximum values of Mode 1. (**F**) Distribution of the CI domain angle in the SC (in *blue*) and CI (in *orange*). (**G, J**) Conformational changes in the Rieske subunit between SC (in *blue*) and CIII$_2$ (in *green*) affect (**G**) the QH$_2$ binding in the proximal Q$_o$ site and (**J**) the Rieske FeS-heme $c_1$ distance in the distal monomer. (**H**) Distribution of the dihedral angle in TM3 of ND6. (**I**) Overview of the SC showing the locations of ND6 and Rieske subunits, as well as the proximal and distal protomers of CIII$_2$.

the SC, that results in a larger *twisting* angle relative to the isolated CI (*Figure 3F*). These alternations suggest that the *opening/closing* and *twisting* motions could be sterically hindered by the adjacent CIII$_2$ (see *Videos 1–5*), and that the *active/deactive* (A/D) transition is coupled to the global motions of the SC. Consistent with our previous findings (*Jussupow et al., 2019*), CI also shows another dominant *bending* motion, where the membrane domain rotates relative to the hydrophilic domain within the membrane plane.

CIII$_2$ also undergoes dynamical changes upon the SC formation, particularly by enhancing the normal modes connected to the *opening/closing* transitions around the iron–sulfur (FeS) Rieske center

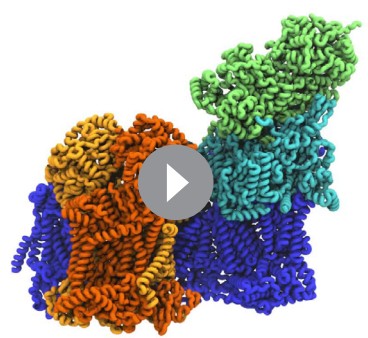

**1: Breathing Motion**

**1: Opening/Closing**

**Video 1.** Normal modes of the SC.
https://elifesciences.org/articles/102104/figures#video1

**Video 2.** Normal modes of the isolated CI.
https://elifesciences.org/articles/102104/figures#video2

(*Figure 3B, E*) that could affect the electron transfer activity of $CIII_2$ (*Crofts, 2004*). In the $SCI/III_2$, $QH_2$ binds further away from the FeS center in the proximal $Q_o$ site, while in the distal site, the FeS center moves closer to the heme $c_1$ (*Figure 3G, I, J*). This asymmetry suggests that the global motion of the SC could regulate the 'preferred' $Q_o$ site for the electron bifurcation process and possibly favor the electron transfer onwards to the Complex IV (CIV) that resides on the distal side of the $CIII_2$ module in the respirasome ($SCI/III_2/IV$) (*Gu et al., 2016*; *Wu et al., 2016*; *Letts et al., 2016*) (cf. also graph theoretical analysis, *Appendix 1—figure 11C, D*). Taken together, these effects suggest that the dynamics of CI and $CIII_2$ show some correlation that could result in allosteric effects, as also suggested by a cryo-EM study of the mitochondrial SC (*Letts et al., 2019*). In this regard, we find that the ligand state of CI (*apo* or $QH_2$) affects the conformational dynamics and the interaction interface of the $SCI/CIII_2$ (*Appendix 1—figure 12A–E*). This surprising long-range effect is likely to result from the increased flexibility of the *apo* state (*Appendix 1—figure 12G*) that, in turn, modulates the interaction at the interface of the SC. Interestingly, similar ligand-dependent conformational changes affecting both the CI and $CIII_2$ domains of the SC are also supported by recent in situ cryo-EM structures of mitochondrial SCs (*Zheng et al., 2024*).

## Enthalpy–entropy compensation drives SC formation

Our molecular simulations suggest that while the membrane strain provides a thermodynamic driving force for the SC formation, the molecular interactions at the interface of the assembly are essential for enthalpically stabilizing the SC over non-specific protein assemblies (*Appendix 1—figure 12*). To assess how the SC formation is affected by the strain effects, protein–protein interactions, and the protein/lipid ratio, we developed a simple statistical-mechanical lattice model of the CI and $CIII_2$ diffusion in the IMM (*Figure 4A*). To this end, the protein–protein interactions were described by tunable interaction energies ($E_{specific}$ and $E_{non-specific}$), modeling the specific hydrogen-bond/salt-bridge interactions and the non-specific interactions, while the membrane–protein interactions were tuned by the strain

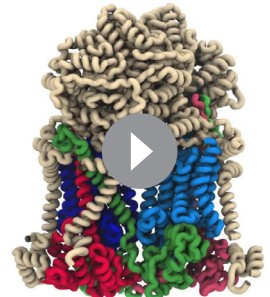

**1: Hinge 1**

**Video 3.** Normal modes of the isolated $CIII_2$.
https://elifesciences.org/articles/102104/figures#video3

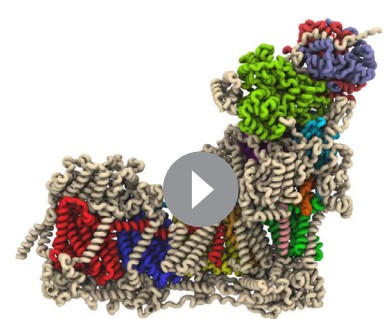

**Mode 1**

**Video 4.** Comparison of the normal modes between the SC and the isolated CI.

https://elifesciences.org/articles/102104/figures#video4

energy ($E_{strain}$). Our model suggests that the ratio between the strain energy and specific interactions indeed provides the key driving force for the SC formation, with a small (30%) decrease in $E_{strain}$ leading to a ca. 60% decrease in the SC population (*Figure 4B*). Our model further predicts that the SC population drastically increases at a specific interaction energy threshold ($E_{specific} < -2 k_B T$) relative to the strain contribution (*Figure 4B*), while the number of non-specific assemblies (adjacent proteins in non-SC orientations) relative to SCs is determined by the ratio of $E_{specific}$ to $E_{non\text{-}specific}$. With a high membrane strain, we find that an increase in temperature also favors the formation of SC assemblies, as it is entropically favored to reduce the overall number of ordered (strained) lipid molecules around the OXPHOS proteins (*Appendix 1—figure 15*). Similarly, a high *protein-to-lipid* ratio significantly increases the population of SCs, suggesting that a high protein packing in IMMs favors the SC formation, while the strength of specific interactions determines the relative population of non-specific assemblies in crowded environments. Taken together, our lattice model, despite its simplicity, captures and validates key features observed in our molecular simulations and supports that the SC formation is affected by both enthalpic and entropic effects.

## Discussion

We have shown here that the respiratory chain complexes perturb the IMM by affecting the local membrane dynamics. The perturbed thickness and alteration in the lipid dynamics lead to an energetic penalty, which can be related to molecular strain effects, as suggested by the changes of both the internal energy of lipid and their interaction with the surroundings (*Appendix 1—figures 2, 5, and 6*), which are likely to be of enthalpic origin. However, lipid binding to the OXPHOS complex also results in a reduction in the translational and rotational motion of the lipids and quinone (*Appendix 1—figures 9 and 10*), which could result in entropic changes. The strain effects are therefore likely to arise from a combination of enthalpic and entropic effects. In this regard, we suggest that the SC assemblies form by condensation of local high-entropic membrane regions around the OXPHOS proteins (*Figure 5A*). However, as the entropic effects also favor the formation of non-specific protein assemblies, unique interactions, such as the ion-paired network around UQCRC1 and NDUFB9/NDUFB4 (*Figure 3A*), are likely to enthalpically stabilize the SC assemblies. The suggested principles show similarities to the hydrophobic effect driving protein folding by condensation of locally ordered water clusters around unfolded protein patches (*Chandler, 2005*).

We note that the magnitude of the estimated bending energies (~$10^2$ kcal mol$^{-1}$) (*Appendix 1—table 1*), while seemingly high at first glance, falls within the range expected for large-scale membrane deformation processes induced by large multi-domain proteins. For example, the Piezo mechanosensitive channel performs roughly

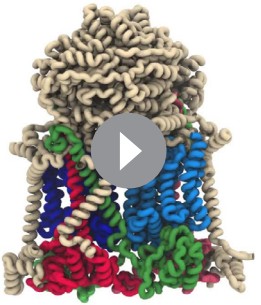

**Mode 1**

**Video 5.** Comparison of the normal modes between the SC and the isolated CIII$_2$.

https://elifesciences.org/articles/102104/figures#video5

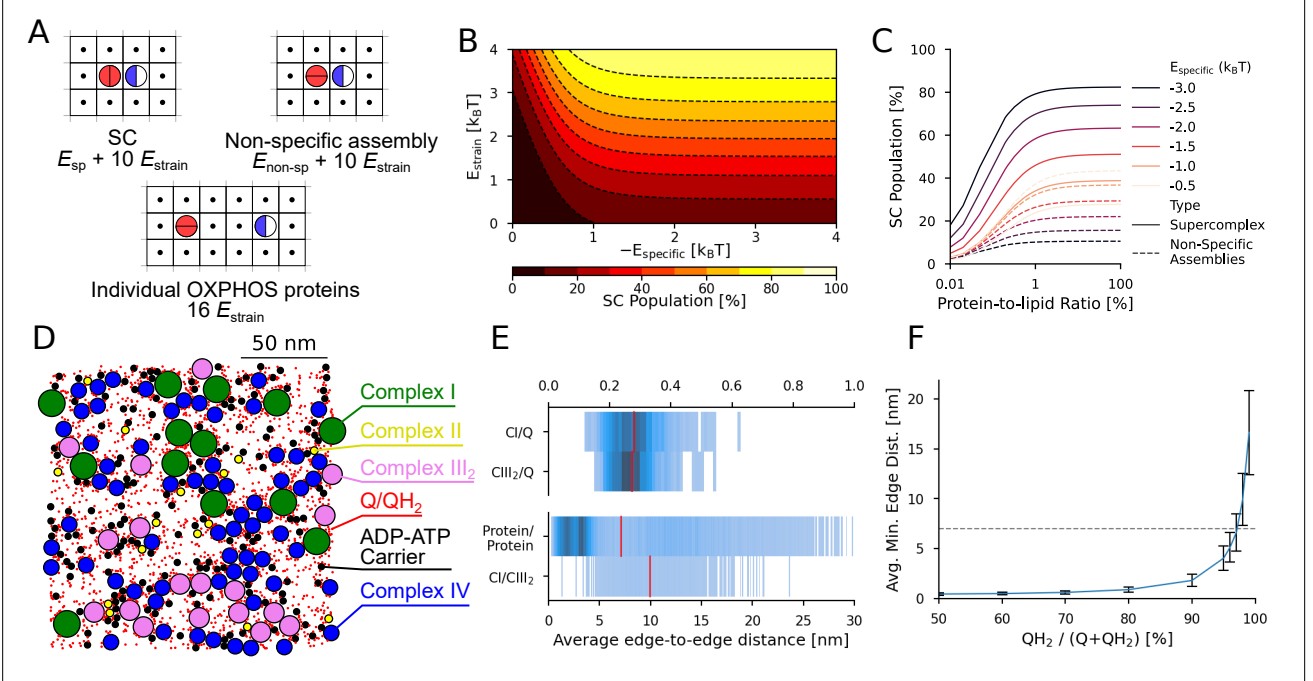

**Figure 4.** Lattice model of SC formation and crowding effects in the IMM. (**A**) Possible specific and non-specific interactions between CI (blue/white) and CIII$_2$ (red) and their respective energies in the lattice model. (**B**) SC population as a function of the specific interaction energy ($E_{specific}$) and molecular strain ($E_{strain}$). (**C**) Specific and non-specific assemblies as a function of the protein–lipid ratio with varying specific interaction energies ($E_{specific}$). (**D**) Representative protein arrangement in crowded IMMs. (**E**) Average *edge-to-edge* distance distributions and nearest neighbor distance (red line) for specific protein–Q/protein contacts. (**F**) Nearest neighbor distance between Q and CI as a function of the Q/QH$_2$ ratio. The dashed line indicates the 7 nm distance between active sites in the SC.

150 $k_BT$ (≈ 90 kcal mol$^{-1}$) of work to bend the bilayer into its dome-like shape (***Guo and MacKinnon, 2017***). Comparable energies have also been estimated for the nucleation of small membrane pores (***Aeffner et al., 2012***), while vesicle formation typically requires bending energies on the order of 300 kcal mol$^{-1}$, largely independent of vesicle size (***Kozlov et al., 2014***). When normalized by the affected membrane area (~1000 nm$^2$), these values correspond to an energy density of approximately 0.1 kcal mol$^{-1}$ nm$^{-2}$, which places our estimates within a biophysically reasonable regime. Notably, cryo-EM structures of supercomplexes show that such assemblies can impose significant curvature on the surrounding bilayer (***Mühleip et al., 2023***; ***Zheng et al., 2024***; ***Waltz et al., 2025***), supporting the notion that respiratory chain organization is closely coupled to local membrane deformation. Nevertheless, we expect that the absolute deformation energies may be overestimated, as the continuum Helfrich model neglects molecular-level effects such as lipid tilt and local rearrangements, which can partially relax curvature stresses and reduce the effective bending penalty near protein–membrane interfaces (***Ergüder and Deserno, 2021***; ***Fiorin et al., 2023***).

We further probed the thermodynamic effects underlying the SC formation by developing a 2D lattice model. Despite its simplicity, the model supports that the SC stability is determined by a delicate balance between membrane strain effects and specific protein–protein interactions, but also strongly affected by the protein concentration and temperature effects (***Figure 4B***). ***Moreno-Loshuertos et al., 2023*** recently suggested that elevated temperatures (>43°C) may indeed disrupt SCs, although the stability of the individual OXPHOS proteins was also decreased in the studied conditions. As the enthalpic effects are of electrostatic origin, the SC stability could also be sensitive to the ionic strength and the PMF, which in turn depends on the metabolic state of the mitochondria.

At the molecular level, the SC formation leads to an accumulation of CDL at the protein–membrane interface (***Figure 2N***), as well as a local Q/QH$_2$ pool near the substrate channel of CI and CIII$_2$ (***Figure 2M***). We find that CDL prefers thinner membranes relative to the neutral phospholipids (PE/PC, ***Appendix 1—figure 5E***), and could thus partially compensate for the hydrophobic mismatch between the OXPHOS proteins and the membrane (***Appendix 1—figure 1***). The Q diffusion is affected

by both specific interactions with OXPHOS proteins and the local membrane thickness (*Figure 2I*). The CDL around the SC is thus likely to have both structural and functional consequences, consistent with its effect on the activity and dynamics of several membrane proteins (*Fry and Green, 1980*; *Fry and Green, 1981*; *Jiang et al., 2000*; *Acehan et al., 2011*, cf. also *Jussupow et al., 2019*). In this regard, CDL was suggested to enhance the substrate dynamics within the Q-tunnel of CI that requires a *twisting–bending* motion around the membrane and hydrophilic domains (*Jussupow et al., 2019*), while destabilization of CDL-binding sites has indeed been shown to disrupt SCs, for example, the $SCIII_2IV_1$ in yeast (*Berndtsson et al., 2020*). Moreover, defects in the CDL synthesis, for example, in the Barth syndrome (*McKenzie et al., 2006*), result in the disassembly of SC, indirectly supporting the involvement of CDL as a 'SC glue'. In this regard, electrostatic effects arising from the negatively charged cardiolipin headgroup could play an important role in the interaction of the OXPHOS complexes. While CDL was modeled here in the singly anionic charged state (but *Appendix 1—figure 5E*), we note that the local electrostatic environment could tune their $pK_a$ that results in protonation changes of the lipid, consistent with its function as a proton collecting antenna (*Haines and Dencher, 2002*).

In addition to the changes in the membrane properties, we observed that the SC formation modulates the conformational dynamics of the individual OXPHOS proteins, especially the large-scale *bending–twisting* motion of CI, the conformation of individual TM helices and conserved loops around proton channels in CI, as well as the motion of the Rieske domain in $CIII_2$ (*Figure 3*, *Appendix 1—figure 12*). The dynamics of these regions are likely to modulate the activity of the OXPHOS proteins, for example, the A/D transition of CI (*Babot et al., 2014*; *Vinogradov, 1998*) that regulate the $\Delta$pH-driven quinol oxidation and reverse electron transfer (*Lambert and Brand, 2004*; *Russell et al., 2020*). Indeed, blocking loop motions surrounding these regions inhibits the proton pumping activity of CI (*Cabrera-Orefice et al., 2018*), while the perturbed motion of the Rieske domain could modulate the electron bifurcation in $CIII_2$ and subsequent electron transfer to CIV. In this regard, we suggest the preferred $QH_2$ binding in the distal $Q_o$ site has functional implications for the respirasome ($SCI/III_2/IV$), where the CIV module is located on the distal side of the $CIII_2$ protomer. The changes in the conformational dynamics upon SC formation may thus affect ROS production (*Lenaz et al., 2016*; *Panov et al., 2007*) via the A/D transition of CI (*Babot et al., 2014*), although it should be noted that no differences in ROS generation were observed for mice unable to form SCs relative to WT mice (with ca. 75% SCs) under normal conditions (*Milenkovic et al., 2023*). It is possible that differences occur only under more strained conditions, for example, in hypoxia or together with disease-related mutations in the OXPHOS proteins, where the SCs could become functionally more important (see below).

To understand how SCs could influence the charge currents in the IMMs, we note that the local increase of the quinol concentration near the active site of CI, and the local decreased quinol pool around the proximal $Q_o$ site of $CIII_2$ could create a substrate gradient ($\nabla c$) and affect the quinol flux between CI and $CIII_2$, $J_{CI\rightarrow CIII} = -D_{mem}\nabla c$ ($D_{mem}$ – $Q/QH_2$ diffusion constant) – if the quinol concentration is rate-limiting for the function of CI or $CIII_2$ (but see below). The 2D-diffusion time for the quinol, $\tau = <r^2>/4D_{mem}$, between the OXPHOS complexes depends on the effective protein–protein distance ($r$), which can be estimated from the protein packing density (cf. *Schlame, 2021*; *Davies et al., 2018*; *Sun et al., 2005*; *Zhou et al., 2015*; *Pebay-Peyroula et al., 2003*; *Petrache et al., 2000*; *Morgenstern et al., 2021*). Using the experimental protein copy numbers in IMM (cf. *Schlame, 2021*; *Davies et al., 2018*; *Sun et al., 2005*; *Zhou et al., 2015*; *Pebay-Peyroula et al., 2003*; *Petrache et al., 2000*; *Morgenstern et al., 2021* and Methods, *Appendix 1—table 6*), we obtain average *edge-to-edge* distance between CI and $CIII_2$ of around 12 nm (*Figure 4E*, cf. also *Schlame, 2021*), which can be compared to the *edge-to-edge* distance of ca. 7/12 nm between the Q tunnel of CI and the proximal/distal $Q_o$ sites of $CIII_2$ within the SC. The reduced diffusion distance could thus provide a subtle rate enhancement ($\tau_{SC}(r = 7\text{ nm})/\tau_{nonSC}(r = 12\text{ nm}) \sim 0.3$) in favor of the SCs under substrate-limited conditions (*Figure 4E, F*). We expect that the small variations in the lateral quinol diffusion (0.4 nm ns$^{-1}$ near CI, 0.8 nm ns$^{-1}$ in bulk) around the SC (*Appendix 1—figures 3D–I and 9*) could favor the diffusion along the membrane patches, although this diffusion is also affected by non-specific collisions. However, due to the overall 3–5% $Q/QH_2$ concentration in IMMs, we estimate that each OXPHOS protein is surrounded by around 6 quinone/quinol species depending on the respiratory conditions (*Figure 5*, see Methods), and leading to a small (0.3 nm) nearest neighbor distance between Q and $CI/CIII_2$ (*Figure 4E*). This implies that each OXPHOS protein has a saturated substrate Q pool within

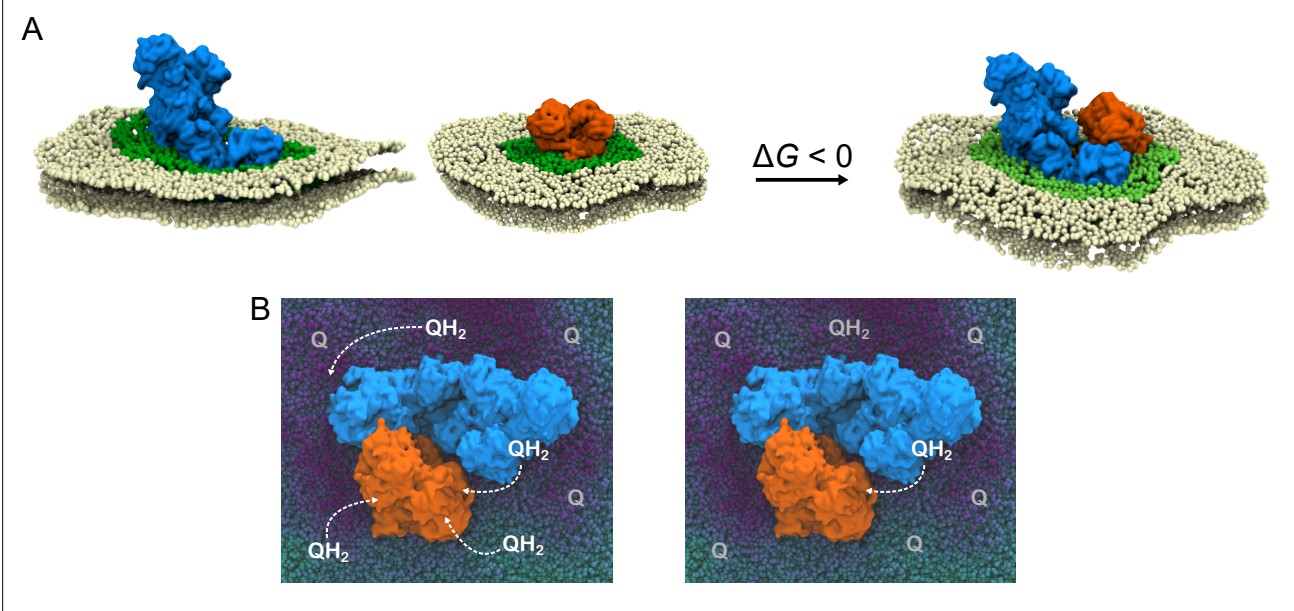

**Figure 5.** Proposed thermodynamic and functional effects of SCs. (**A**) The individual OXPHOS complexes, CI and CIII$_2$, induce prominent molecular strain in the surrounding lipid membrane (dark green area) due to a hydrophobic mismatch between the protein and the membrane. Condensation of locally strained membrane patches entropically drives the SC formation and leads to an overall reduction of the membrane strain (light green area around the SC), favoring the accumulation of cardiolipin around the SC. (**B**) During normal respiratory conditions, each OXPHOS complex is surrounded by multiple (ca. 6) quinol/quinone molecules that can act as substrates for the proteins. At limiting QH$_2$ concentration, the quinol diffusion between CI and CIII$_2$ becomes rate-limiting and leads to a kinetic advantage of the SC (see also *Figure 4F*).

its 'reaction sphere' upon high charge flux conditions (here 50% reduced Q pool). In other words, the CIII$_2$ node of the SC does not need to rely on the quinol generated at the CI node, but can instead oxidize a quinol molecule from the surrounding Q pool. In this regard, *Fedor and Hirst, 2018* found that the alternative oxidase (AOX) co-reconstituted into liposomes outcompetes the quinol re-oxidation rate of the SCI/III$_2$, suggesting that the diffusion of the Q-species between the OXPHOS complexes is not rate-limiting under the studied conditions. However, at low QH$_2$/Q ratios (<5% in the model), the minimal nearest neighbor distance between the Q pool and CI/CIII$_2$ drastically increases (*Figure 4D–F*), which in turn leads to a kinetic preference for the SC (*Figure 4D*).

At the microscopic level, the SC influences the lipid and water dynamics at the protein–lipid interface and the physico-chemical properties of the IMM, including its local dielectric properties of the membrane. It is possible that such local microenvironments have functional implications that could affect the proton conduction along the membrane, with important bioenergetic consequences. Taken together, our combined findings suggest that SC forms as a result of a complex interplay between molecular interactions and membrane strain effects that control the functional dynamics of the OXPHOS proteins.

## Materials and methods
### aMD simulations

aMD simulations of the ovine Complex I (CI), Complex III$_2$ (CIII$_2$), and the I–III$_2$ supercomplex (SCI/III$_2$) were performed in a POPC:POPE:cardiolipin (2:2:1) membrane containing 5 mol% QH$_2$/Q (1:1 ratio). Cardiolipin was modeled as tetraoleoyl cardiolipin (18:1,18:1,18:1,18:1) with a headgroup modeled in a singly protonated state (with Q$_{tot}$ = −1). The system was solvated with TIP3P water molecules and 150 mM NaCl. The simulations were performed in different ligand states (*apo*, Q/QH$_2$ bound, *Appendix 1—table 2*) at T = 310 K and P = 1 bar using NAMD2.14 (*Phillips et al., 2020*) with a 2-fs integration timestep and long-range electrostatic interactions treated using the Particle Mesh Ewald approach. During construction of the simulation setups, it was carefully considered that no leaflet introduced higher lipid densities that could result in artificial displacement effects. The systems

comprised 0.8–1.65 million atoms and were modeled using the CHARMM36 force field (*Best et al., 2012*) in combination with in-house DFT-based parameters (*Vanommeslaeghe et al., 2010*) of the co-factors. Protonation states were established based on electrostatic calculations with Monte Carlo (MC) sampling techniques using APBS/Karlsberg+ (*Baker et al., 2001*; *Kieseritzky and Knapp, 2008a*; *Kieseritzky and Knapp, 2008b*).

MD simulations of the ovine CI were conducted based on a cryo-EM structure (PDB ID: 6ZKC *Kampjut and Sazanov, 2020*), with simulations performed for both the quinol-bound and *apo* forms. Unresolved regions in the supernumerary subunits NDUFA7 and NDUFB6 were modeled using Colab-Fold (*Mirdita et al., 2022*). The system was equilibrated for 100 ns in the $QH_2$-bound state, which was then used to propagate both states for $2 \times 0.5$ µs each.

The ovine Complex $III_2$ was modeled based on the cryo-EM structure (PDB ID: 6Q9E *Letts et al., 2019*), with simulations also performed in different ligand states (*Appendix 1—table 2*). After equilibration of the system for 100 ns with the distal and proximal Q sites occupied, the system was propagated in each state for $2 \times 0.5$ µs.

A fully atomistic model of the SC was constructed based on the high-resolution structures of CI and $CIII_2$ that were merged into the experimental structure of the supercomplex (PDB ID: 6QBX *Letts et al., 2019*). After equilibration of the system for 100 ns with all Q sites occupied in CI and $CIII_2$, the complete system was propagated for $2 \times 0.5$ µs in each state (see *Appendix 1—table 2*).

Membrane systems with 1:1 POPC/POPE or CDL were constructed using CharmmGUI with a membrane area of $80 \times 80$ Å$^2$, a hydration layer of 45 Å, and NaCl or KCl concentrations of 150 mM. See *Appendix 1—table 3* for a detailed description of the membrane compositions.

## cgMD simulations

cgMD simulation models of the ovine CI, $CIII_2$, and the $SCI_1/III_2$ were created based on the atomistic models using the MARTINI3 force field (*Souza et al., 2021*) and Gromacs (*Abraham et al., 2015*). All simulations were constructed with identical amounts of lipid molecules, a 2:2:1 POPC:POPE:cardiolipin ratio, and a 3 mol% mixture of quinone/quinol embedded in a simulation system with dimensions $47 \times 47 \times 31$ nm$^3$ (for the SC, see *Appendix 1—table 2*). All simulations were carried out at 310 K in an *NPT* ensemble using the velocity rescaling thermostat (*Woodcock, 1971*; *Bussi et al., 2007*), the Parrinello–Rahman barostat (*Parrinello and Rahman, 1981*) and a 20 fs timestep using GROMACS (*Abraham et al., 2015*). The protein structures were stabilized with elastic networks on backbone beads using a cutoff distance of 0.9 nm with a force constant of 500 kJ mol$^{-1}$ nm$^{-2}$. The elastic network was also applied between residues of different subunits, but not between CI and $CIII_2$, while cgMD parameters for all cofactors were also developed. Two replicas were run for CI and $CIII_2$ ($2 \times 50$ µs each), and for the $SCI/III_2$ ($2 \times 50$ µs), as well as 23 µs cgMD simulations of the membrane with the same number of lipid and $Q/QH_2$ molecules as in the protein simulations. Additional 5 µs cgMD simulations of the membrane systems, as well as for all protein models, were performed with longer lipids (0.44 nm, 0.50 nm, and 0.53 nm instead of 0.47 nm).

## Lattice model of SC formation

A lattice model of the CI and $CIII_2$ was constructed (*Figure 4A, B*) by modeling the OXPHOS proteins in unique grid positions on a 2D $N \times N$ lattice, with $N = [6,15]$. Depending on the relative orientation, the protein–protein interaction was described by specific interactions (giving rise to the energetic contribution $E_{\text{specific}} < 0$) and non-specific interactions ($E_{\text{non-specific}} > 0$), whereas the membrane–protein interaction determined the strain energy of the membrane ($E_{\text{strain}}$), based on the number of neighboring 'lipid' occupied grids that are in contact with proteins (*Figure 4A*). The interaction between the lipids was indirectly accounted for by the background energy of the model. The proteins could occupy four unique orientations on a grid ($g_i = [North, East, South, West]$). The states and their respective energies that the system can visit are summarized in *Appendix 1—table 7*.

The total Hamiltonian of the lattice model can be written as,

$$H = \sum_{i,j}^{N,N} g_i g_j E_{\text{specific}} + \sum_{i,j}^{N,N} d_i d_j E_{\text{non-specific}} + \sum_{i,j}^{N,N} \delta_{i,j} E_{\text{strain}}, \quad (1)$$

with

$$g_i g_j = \begin{cases} 1, & \text{for } i = \text{left}, j = i + 1 = \text{right}, \\ 0, & \text{otherwise} \end{cases} \tag{2}$$

$$d_i d_j = \begin{cases} 0, & i = \text{left}, j = i + 1 = \text{right}, \\ 1, & \text{otherwise} \end{cases} \tag{3}$$

$$\delta_{i,j} = \begin{cases} 1, & \text{for unoccupied } j = i + 1 \\ 0, & \text{otherwise} \end{cases} . \tag{4}$$

The $g_i g_j$-term assigns a specific energy contribution when the OXPHOS complexes are in adjacent lattice points only in a correct orientation (modeling a specific non-covalent interaction between the complexes such as the Arg29[FB4]–Asp260[C1]/Glu259[C1] interaction between CI and CIII$_2$). The $d_i d_j$-term assigns a non-specific interaction for the OXPHOS complexes when they are in adjacent lattice points, but in a 'wrong' orientation relative to each other to form a specific interaction. The $\delta_{ij}$ term introduces a strain into all lattice points surrounding an OXPHOS complex, mimicking the local membrane perturbation effects observed in our molecular simulations.

This leads to the partition function,

$$Z = \sum_{i,j=1}^{N,N} w_i e^{-\beta E_{i,tot}}, \tag{5}$$

where $w_i$ is the degeneracy of the state, modeling that the SC and OXPHOS proteins can reside at any lattice position of the membrane, and where $\beta = 1/k_B T$ ($k_B$, Boltzmann's constant; $T$, temperature). The probability of a given state $i$ was calculated as,

$$p_i = \frac{w_i e^{-\beta E_{i,tot}}}{Z}, \tag{6}$$

with the free energy ($G$) defined as,

$$G = -\beta^{-1} \ln Z. \tag{7}$$

The conformational landscape was sampled by MC using $10^7$ MC iterations with 100 replicas. Temperature effects were modeled by varying $\beta$, and the effect of different *protein-to-lipid* ratios by increasing the grid area. The simulation details can be found in *Appendix 1—table 8*.

## Statistical model of the membrane distribution in the IMM

Based on the experimental protein copy numbers (cf. *Schlame, 2021*) and average protein areas, the proteins in the IMM were modeled as randomly positioned circles on a membrane square with a side length of 163 nm (see *Appendix 1—table 6*). The system was minimized according to the Hamiltonian,

$$H = \frac{1}{2} \sum_i \sum_{j \neq i} f_s (d_{ij}), \tag{8}$$

$$f_s (d_{ij}) = \begin{cases} k \cdot (r_i + r_j - d_{ij}), \text{ if } d_{ij} < r_i + r_j \\ 0, \text{ otherwise} \end{cases}, \tag{9}$$

where $d_{ij}$ is the distance between two proteins $i$ and $j$ with respective radii, $r_i$ and $r_j$, while the force constant $k$ was introduced to avoid steric clashes and set to 1 kcal mol$^{-1}$ nm$^{-1}$. This Hamiltonian corresponds to a two-dimensional system consisting of rigid, non-deformable circular particles.

## Estimation of membrane strain

The potential of mean force of the membrane thickness profile ($z$) was calculated based on,

$$G_{\text{memb}}(z) = -k_B T \ln \rho(z), \tag{10}$$

where $z$ is the membrane thickness and $\rho$ is the probability distribution of the membrane thickness. The membrane strain induced by the protein was calculated from the cgMD simulations of the different systems as (see Methods),

$$G_{\text{strain}}^{\text{protein}} = \int \rho^{\text{protein}}(z) \, G_{\text{memb}}(z) \, \mathrm{d}z. \tag{11}$$

To obtain detailed atomistic insight into the strain effects, the local lipid strain was quantified by energy decomposition analysis, by comparing the internal lipid energies to the average lipid energy from a membrane simulation (*Appendix 1—figure 5*). The simulation data were averaged over 690,000 datapoints from 1.85 μs MD simulations of systems A1,7,9 and M4 (*Appendix 1—tables 2 and 3*).

## Estimation of dielectric profiles

The dielectric constant ($\varepsilon$) was determined from the local variance of the dipole moment at a given position $\boldsymbol{r}_p$ using the Kirkwood–Fröhlich equation (*Kirkwood, 1939*, *Fröhlich, 1958*),

$$\varepsilon = 1 + \frac{\text{var}\,\boldsymbol{M}}{3\varepsilon_0 k_B T V}, \tag{12}$$

with the local dipole moment calculated from the sum of the charge-weighted positions of all atoms within 0.2 nm of $\boldsymbol{r}_p$,

$$\boldsymbol{M}(\boldsymbol{r}_p, t) = \sum_{i,\, |r_i - r_p| < 0.2nm} q_i \boldsymbol{r}_i, \tag{13}$$

where $\varepsilon_0$ is the vacuum permittivity, $k_B$ is the Boltzmann constant, $T$ is the temperature, and $V$ is the volume of the selection sphere. The calculations were performed by averaging the total **M** over fixed $z$ values from the membrane plane. Note that this treatment differs from extraction of radial and axial contributions of the dielectric tensor, as developed by Netz and co-workers (cf. *Loche et al., 2019* and refs therein) that requires a more elaborate treatment, which is outside the scope of the present work.

## Analysis of lipid chain *end-to-end* length

To probe the protein-induced deformation effect of the membrane, the membrane curvature ($H$), and the *end-to-end* distance between the lipid chains were computed based on aMD and cgMD simulations. The lipid chain length was computed from simulations A1–A6 and C1 based on the first and last carbon atoms of each lipid chain. For example, the *end-to-end* length of a cardiolipin chain was determined as the distance between atom 'CA1' and atom 'CA18'.

## Membrane curvature and deformation energy

The local mean curvature of the membrane midplane was computed using the Helfrich model (*Helfrich, 1973*; *Campelo et al., 2014*) by approximating the membrane surface as a height function $Z(x,y)$, defined as the average location of the N- and P-side leaflets at each grid point. Based on this, the mean curvature $H(x,y)$ was calculated as,

$$H(x,y) = \frac{\left(1 + Z_x^2\right) Z_{yy} + \left(1 + Z_y^2\right) Z_{xx} + 2Z_x Z_y Z_{xy}}{2\left(1 + Z_x^2 + Z_y^2\right)^{3/2}}, \tag{14}$$

where the derivatives are defined as $Z_i = \frac{\partial Z}{\partial i}$ and $Z_{ij} = \frac{\partial^2 Z}{\partial i \partial j}$.

The thickness deformation energy was computed from the local thickness $d(x,y)$ relative to a reference thickness distribution $F(d)$, derived from membrane-only simulations, and converted to a free energy profile via Boltzmann inversion. At each grid point, the $F(d)$ was summed over the grid,

$$G_{thick} = \sum_{x,y} F\left(d\left(x,y\right)\right) \Delta A$$

(15)

The bending deformation energy was computed from the mean curvature field $H(x,y)$, assuming a constant bilayer bending modulus $\kappa$ (taken as 20 $k_B T$ = 11.85 kcal mol$^{-1}$ *Equation 16*):

$$G_{curv} = \frac{1}{2}\kappa \sum_{x,y} \left(2H\left(x,y\right)\right)^2 \Delta A,$$

(16)

where $\Delta A$ is the area of the grid cell.

The thickness and curvature fields were obtained by projecting the coarse-grained MD trajectories (one frame per ns) onto a 2D grid with a resolution of 0.5 nm. Grid points with low occupancy were down-weighted to mitigate noise. More specifically, points with counts below 50% of the median grid count were scaled linearly by their relative count value. To focus the analysis on the region around the protein–membrane interface, only grid points within a radius of 20 nm from the center of the complex were included in the energy calculations. Energies were normalized to an effective membrane area of 1000 nm$^2$ to facilitate the comparison between systems. Bootstrapping with resampling over frames was performed to estimate the standard deviations of $G_{thick}$ and $G_{curv}$.

We find that $G_{curv}$ converges slowly due to its sensitivity to local derivatives and the small grid size required to resolve the curvature contribution near the protein. Consequently, tens of microseconds of simulations were necessary to obtain well-converged estimates of the curvature energy.

### Spatial integration of cryo-EM maps

The properties of the local membrane environment were estimated based on a high-resolution structure of Complex I (PDB ID: 6RFR, EMD-4873) and the associated cryo-EM map (*Parey et al., 2019*). To this end, we first subtracted the cryo-EM density close to the resolved protein structure, while the density vertical to the membrane plane was approximated as a double sigmoid function. To identify the membrane regions, we used a neural network model (multi-layer perceptron classifier) with 3 × 100 hidden layers that we trained to differentiate between double sigmoid-type regions and regions with different electron density distributions. To this end, we created 20,000 random double-sigmoid distributions with random errors as well as 120,000 of various random distributions. We used 25% of the data as test data, resulting in an accuracy score of 97%.

### Estimation of effective Q/QH$_2$ concentrations

The IMM has an area of around 16,905 nm$^2$ and it comprises 48,300 lipid molecules (*Appendix 1—table 6*). For a 1% Q/QH$_2$ concentration, this implies 483 Q/QH$_2$ molecules. Based on protein copy numbers and the structure of the membrane proteins, the IMM comprises 291 proteins with a total area of 14,586 nm$^2$ (*Appendix 1—table 6*), thus suggesting that two mitochondrial proteins occupy an average area of 216.4 nm$^2$. For a square membrane patch, this would lead to an edge of $R$ = 14.7 nm, and an average protein–protein distance of,

$$
\begin{aligned}
\frac{\langle r_{p-p} \rangle}{R} &= \int_0^1 \int_0^1 \int_0^1 \int_0^1 \sqrt{\left(x_1^2 - x_2^2\right)^2 + \left(y_1^2 - y_2^2\right)^2}\, dx_1 dy_1 dx_2 dy_2 = 4 \int_0^1 \int_0^1 \sqrt{x^2 + y^2}\, dx dy \\
&= 8 \int_0^{\frac{\pi}{4}} \int_0^{\frac{1}{\cos\theta}} \left(1 - r\sin\theta\right)\left(1 - r\cos\theta\right) r^2 dr d\theta \\
&= \frac{1}{15}\left[2 + \sqrt{2} + 5\ln\left(\sqrt{2} + 2\right)\right] \approx 0.52
\end{aligned}
$$

(17)

or $\langle r_{p-p} \rangle$ = 7.6 nm. For the same membrane patch, we thus have an average of 216.4 nm$^2$ × ((48,300 × 0.01)/16,906.0 nm$^2$) = 6.2 Q/QH$_2$ molecules.

### Multiple sequence alignment

Sequences of mammalian species were aligned using the ClustalOmega (*Sievers et al., 2011*) command line interface of Biopython (*Cock et al., 2009*). Sequences of selected species were visualized using Jalview (*Waterhouse et al., 2009*).

### Allosteric network analysis

Interactions between amino acid residues were modeled as an interaction graph, where each residue was represented by a vertex. Two nodes were connected by an edge, if the Cα atoms of the corresponding amino acid residues were closer than 7.5 Å for more than 50% of the frames of simulations A1–A6 (time step of frames: 1 ns) (*Proctor et al., 2015*). This analysis was carried out for the aMD simulations of the supercomplex, analyzing differences between the Q bound and *apo* states (simulations A1 + A2 + A3 vs. A4 + A5 + A6).

## Acknowledgements

This work was supported by grants from the European Research Council (ERC) under the European Union's Horizon 2020 research and innovation program/grant agreement 715311, the Swedish Research Council (VR), and the Knut and Alice Wallenberg foundation (2019.0251, 2019.0043, 2024.0220), and the Göran Gustafsson Foundation for Research in Natural Sciences and Medicine. We are thankful for computing time provided by the Partnership for Advanced Computing in Europe (PRACE project: pr127) to access Piz Daint hosted by the Swiss National Supercomputing Center (CSCS). This work was also supported by the National Academic Infrastructure for Supercomputing in Sweden (NAISS 2025/1-33, 2025/6-165, 2024/1-28, 2023/1-31, 2023/6-128) and the Swedish National Infrastructure for Computing (SNIC 2022/1-29, 2022/6-190).

## Additional information

### Funding

| Funder | Grant reference number | Author |
| --- | --- | --- |
| Knut och Alice Wallenbergs Stiftelse | 2024.0220 | Ville RI Kaila |
| Knut och Alice Wallenbergs Stiftelse | 2019.0251 | Ville RI Kaila |
| Knut och Alice Wallenbergs Stiftelse | 2019.0043 | Ville RI Kaila |
| Göran Gustafsson Foundation for Research in Natural Sciences and Medicine | | Ville RI Kaila |
| National Academic Infrastructure for Supercomputing in Sweden | 2025/1-33 | Ville RI Kaila |
| National Academic Infrastructure for Supercomputing in Sweden | 2025/6-165 | Ville RI Kaila |
| National Academic Infrastructure for Supercomputing in Sweden | 2024/1-28 | Ville RI Kaila |
| National Academic Infrastructure for Supercomputing in Sweden | 2023/1-31 | Ville RI Kaila |
| National Academic Infrastructure for Supercomputing in Sweden | 2023/6-128 | Ville RI Kaila |

| Funder | Grant reference number | Author |
|---|---|---|
| Swedish National Infrastructure for Computing | SNIC 2022/1-29 | Ville RI Kaila |
| Swedish National Infrastructure for Computing | 2022/6-190 | Ville RI Kaila |
| European Research Council (ERC) | 715311 | Ville RI Kaila |

The funders had no role in study design, data collection, and interpretation, or the decision to submit the work for publication.

## Author contributions

Maximilian C Pöverlein, Software, Formal analysis, Investigation, Visualization, Methodology, Writing – review and editing; Alexander Jussupow, Software, Formal analysis, Validation, Investigation, Visualization, Methodology, Writing – original draft, Writing – review and editing; Hyunho Kim, Formal analysis, Validation, Investigation, Visualization, Methodology, Writing – review and editing; Ville RI Kaila, Conceptualization, Resources, Formal analysis, Supervision, Funding acquisition, Investigation, Methodology, Writing – original draft, Project administration, Writing – review and editing

Reviewer #1 (Public review): https://doi.org/10.7554/eLife.102104.4.sa1
Reviewer #3 (Public review): https://doi.org/10.7554/eLife.102104.4.sa2
Author response https://doi.org/10.7554/eLife.102104.4.sa3

# Additional files

## Supplementary files
MDAR checklist

## Data availability
Data available at https://doi.org/10.5281/zenodo.18399378.

The following dataset was generated:

| Author(s) | Year | Dataset title | Dataset URL | Database and Identifier |
|---|---|---|---|---|
| Kaila V, Kim H, Pöverlein M, Jussupow A | 2026 | Protein-Induced Membrane Strain Drives Supercomplex Formation | https://doi.org/10.5281/zenodo.18399378 | Zenodo, 10.5281/zenodo.18399378 |

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

## Appendix 1

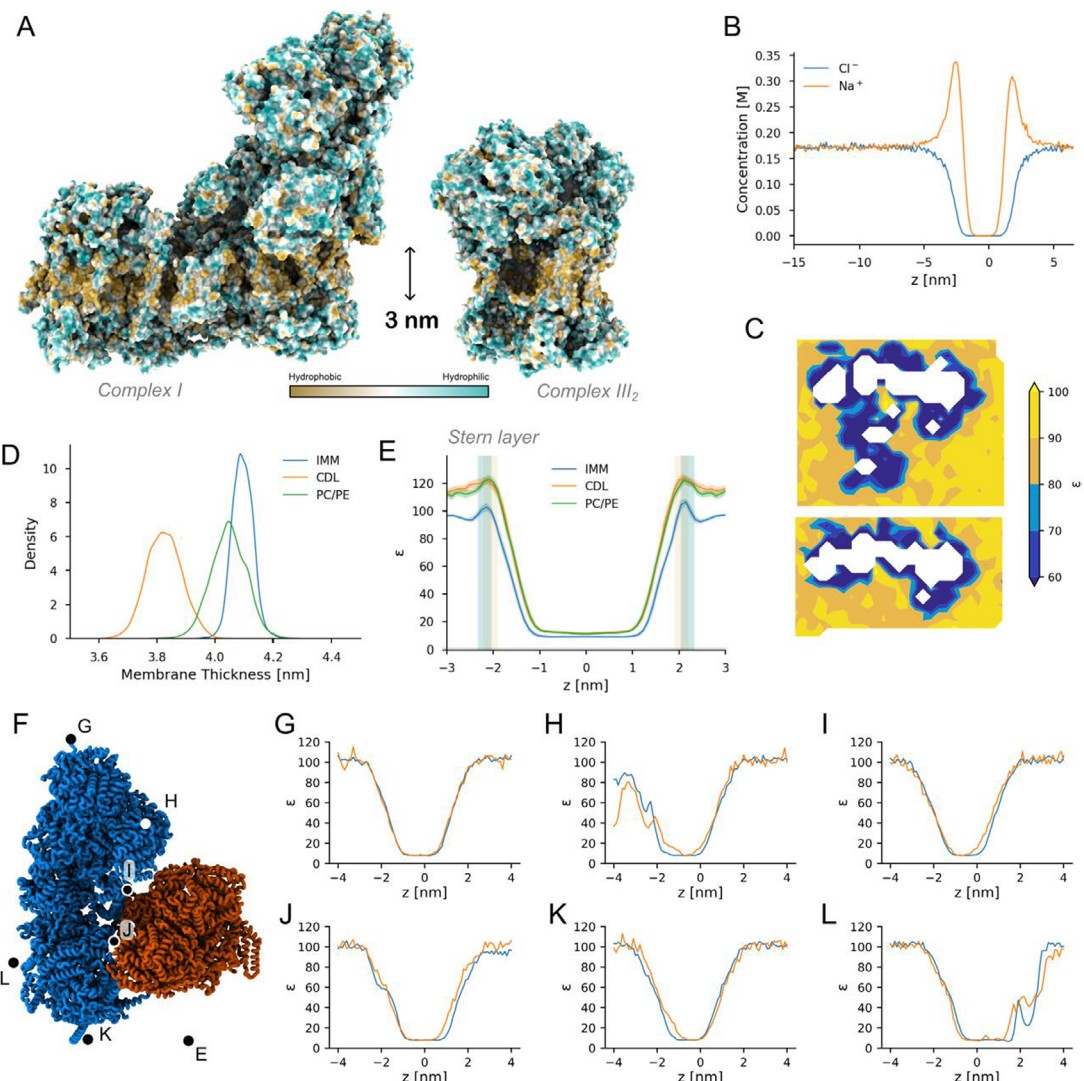

**Appendix 1—figure 1.** Analysis of membrane properties. Mismatch of the hydrophobic region of CI and CIII$_2$ relative to the lipid membrane. (**A**) The hydrophobic region in CI (*left*) and CIII$_2$ (*right*), with hydrophobic amino acids in gold, and hydrophilic amino acids in cyan. The maps were created using ChimeraX. (**B**) Ion distribution as a function of $z$ position in a protein-free slab of simulation S1. (**C**) Dielectric constant (at $z = 0.5$ nm from the membrane–water interface) around the SC (*left*) and CI (*right*), with the SC leading to larger membrane surface with a perturbed $\varepsilon$. (**D**) Distribution of the membrane thickness in a POPC/POPE/CDL/QH$_2$ membrane and a CDL membrane. (**E**) Dielectric profile ($\varepsilon$) computed perpendicular to the membrane plane for a POPC/POPE and a pure CDL membrane. The Stern layer, characterized by the local increase in the $\varepsilon$, extends ca. 1.5 nm from the membrane plane and is affected by the lipid composition. (**F**) Position of sampled dielectric profiles shown in (**G**–**L**).

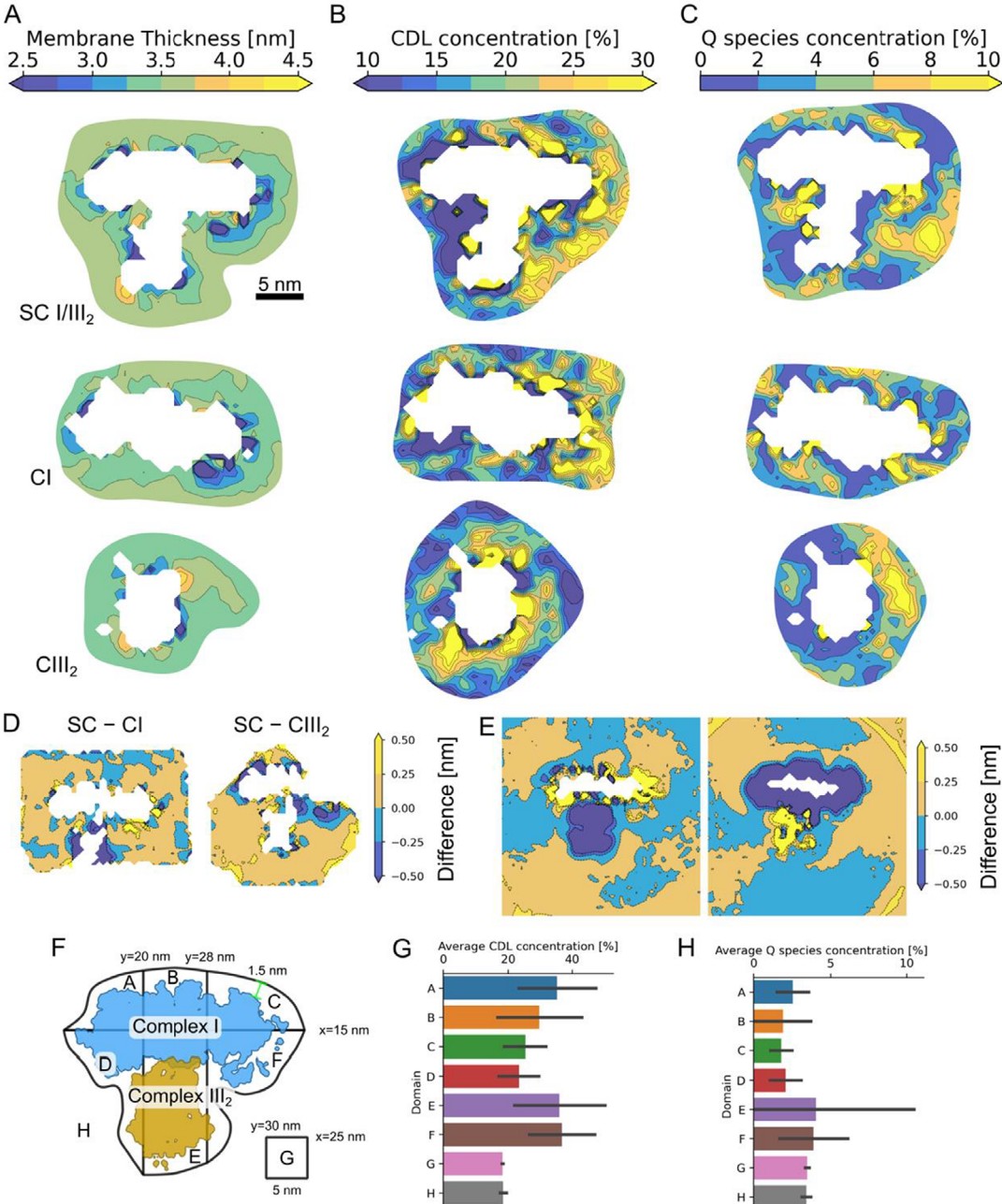

**Appendix 1—figure 2.** Concentration of lipids and quinones, and analysis of membrane thickness in atomistic molecular dynamics (aMD) simulations. (**A**) Local membrane thickness in the SCI/CIII$_2$ (*top*), CI (*middle*), and CIII$_2$ (*bottom*). (**B**) Local cardiolipin concentration. (**C**) Local QH$_2$ concentration. Difference in the membrane thickness around the SC relative to CI (*left*) or relative to CIII$_2$ (*right*) from (**D**) aMD and (**E**) coarse-grained molecular dynamics (cgMD). (**F–H**) Error bars for the local CDL and Q/QH$_2$ enrichment (data presented in *Figure 2H, I*) were estimated by averaging over the membrane in the vicinity of the protein (within 1.5 nm distance) divided into six domains (see panel **F**). The average concentrations for each domain and the calculated errors of (**G**) CDL and (**H**) Q/QH$_2$.

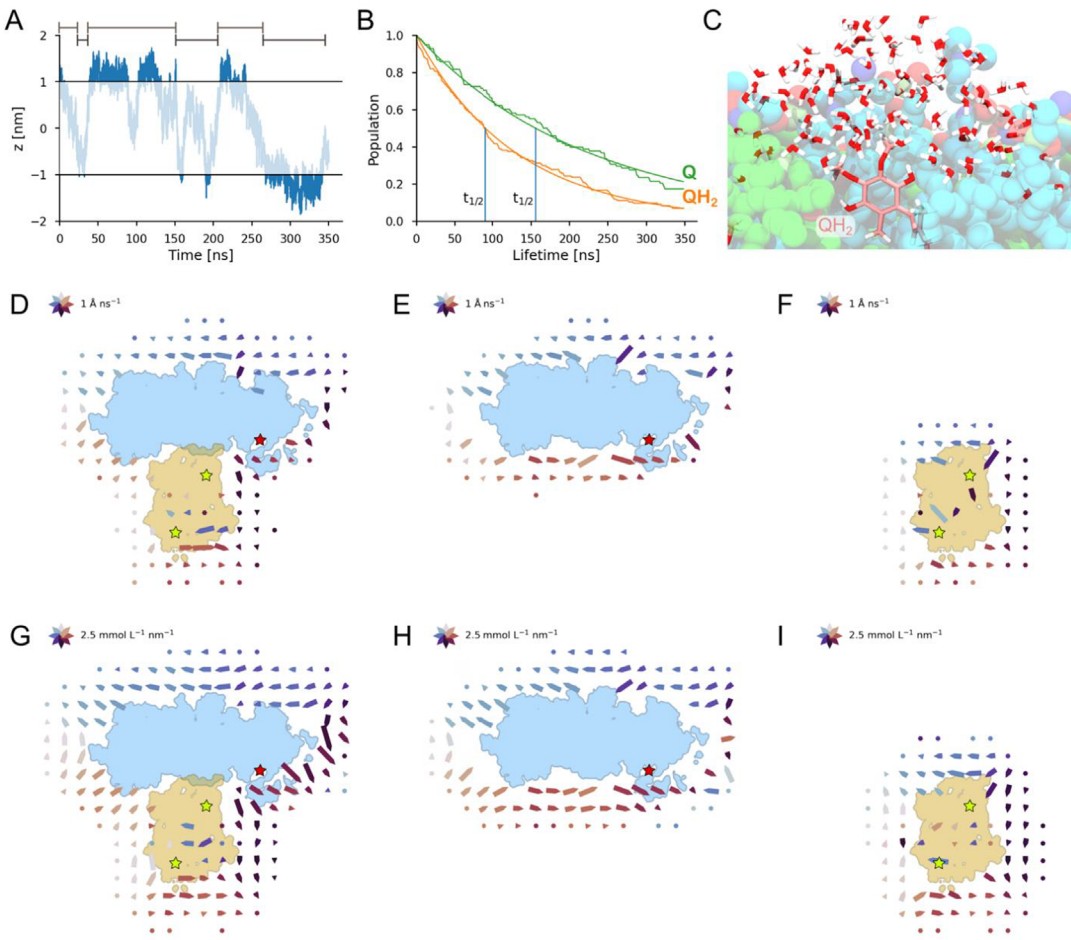

**Appendix 1—figure 3.** Quinone dynamics in the membrane. (**A**) Flip-flop motion of the Q headgroup in the membrane from atomistic molecular dynamics (aMD) simulations. Lifetime spent in upper/lower leaflets indicated by bars above. (**B**) Lifetime analysis of the Q and QH$_2$ headgroup, with the half-life ($t_{1/2}$) for Q/QH$_2$ spent on a given membrane leaflet during consecutive steps during the simulations. (**C**) Representative aMD structure of QH$_2$ headgroup interaction with water and lipid headgroups at the membrane interface. (**D–F**) Average diffusion of Q/QH$_2$ around the SC/III$_2$, CI, and CIII$_2$ from coarse-grained molecular dynamics (cgMD) simulations. (**G–I**) Concentration gradient of Q/QH$_2$ around the SCI/III$_2$, CI, and CIII$_2$ from cgMD simulations. CI – in blue, CIII$_2$ – in beige. The Q species show a directed rotational diffusion around the OXPHOS complexes that could arise from conservation laws due to the Q depletion/increase in unique regions.

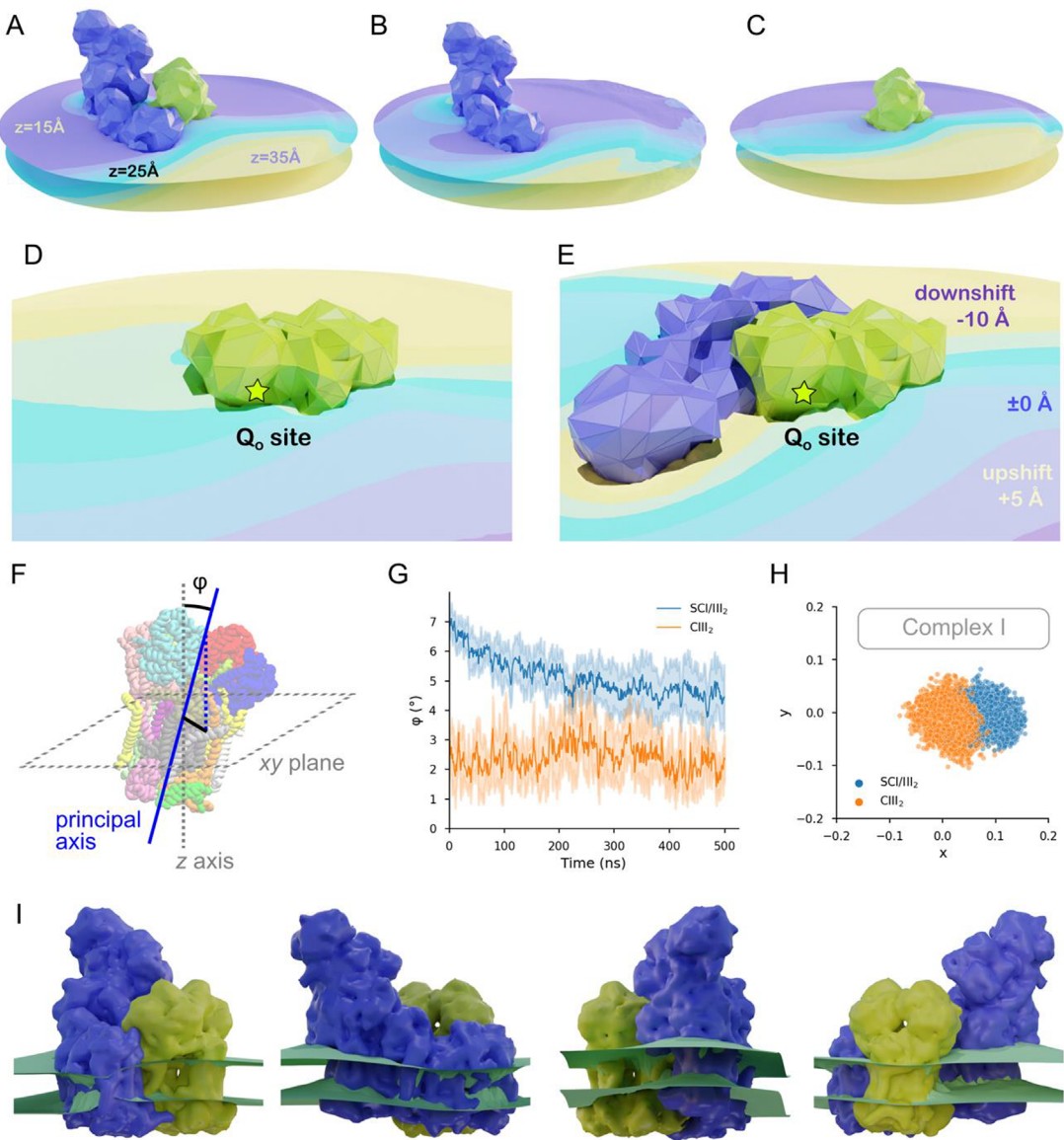

**Appendix 1—figure 4.** Shifts in the membrane leaflet from coarse-grained molecular dynamics (cgMD) simulations. Shifts in the membrane leaflet viewed from the N-side of the (**A**) SCI/III$_2$, (**B**) CI, and (**C**) CIII$_2$. Shifts in the membrane leaflet viewed from the P-side of the membrane for (**D**) CIII$_2$ and (**E**) SCI/III$_2$. The location of the Q$_o$ site of the proximal CIII protomer is indicated by a red star. (**F**) The CIII$_2$ angle was measured between the z-axis of the simulation box and the principal axis of inertia of the protein. (**G**) Time evolution of the membrane angle averaged over atomistic molecular dynamics (aMD) simulations of the SCI/III$_2$ and CIII$_2$. (**H**) Projection of the principal axis onto the xy-plane. (**I**) Visualization of the membrane distortion effect based on trajectory-averaged membrane position.

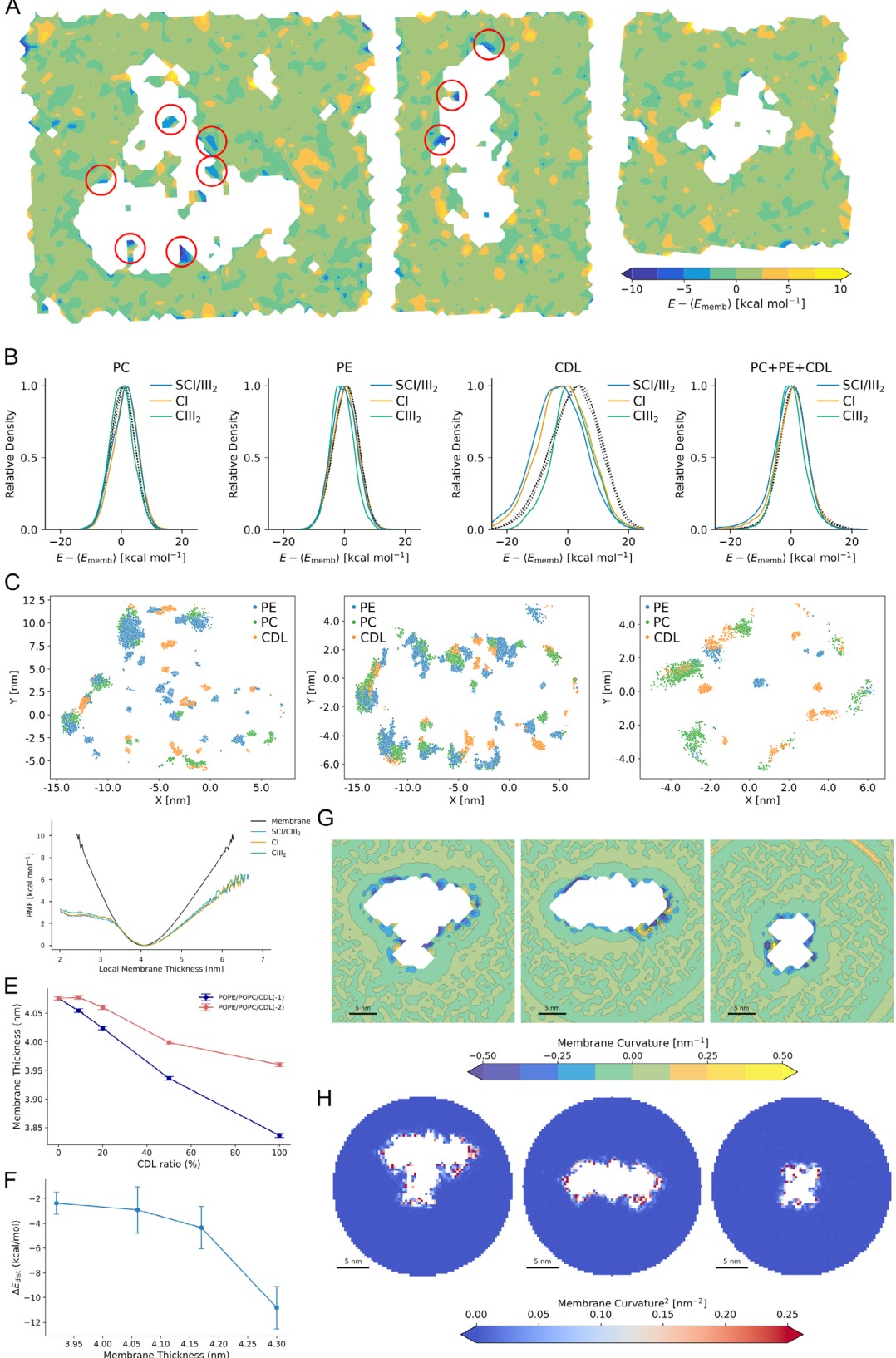

**Appendix 1—figure 5.** Analysis of membrane-induced distortion effects. (**A**) Relative strain effect relative to a lipid membrane from atomistic MD simulations of the SCI/III$_2$, CI, and CIII$_2$, suggesting reduction of the
*Appendix 1—figure 5 continued on next page*

*Appendix 1—figure 5 continued*

membrane strain (blue patches) in the SC surroundings. The figure shows the non-bonded energies relative to the average non-bonded energies from membrane simulations (simulation M4, *Appendix 1—table 2*). (**B**) The lipid strain contribution for different lipids calculated from non-bonded interaction energies of the lipids relative to the average lipid interaction in an IMM membrane model (simulation M4). The figure shows the relative strain contribution for nearby lipids r < 2 Å, in color from panel (**C**), and lipids >5 Å from the OXPHOS proteins. (**C**) Selection of lipids (<2 Å) interacting with the OXPHOS proteins. (**D**) Potential of mean force (PMF) of membrane thickness derived from thickness distributions from coarse-grained molecular dynamics (cgMD) simulations of a membrane, the SCI/III$_2$, CI, and CIII$_2$. (**E**) Membrane thickness as a function of CDL concentration from cgMD simulations. (**F**) $\Delta G_{thick}$ of the SC as a function of membrane thickness based on cgMD simulations. (**G**) Membrane curvature around the SCI/III$_2$ (*left*), CI (*middle*), and CIII$_2$ (*right*) from atomistic simulations. (**H**) Squared membrane curvature obtained from cgMD simulations, within a 20-nm radius around the center of the system. These maps correspond to the curvature field used in the calculation of the bending deformation energy term ($G_{curv}$).

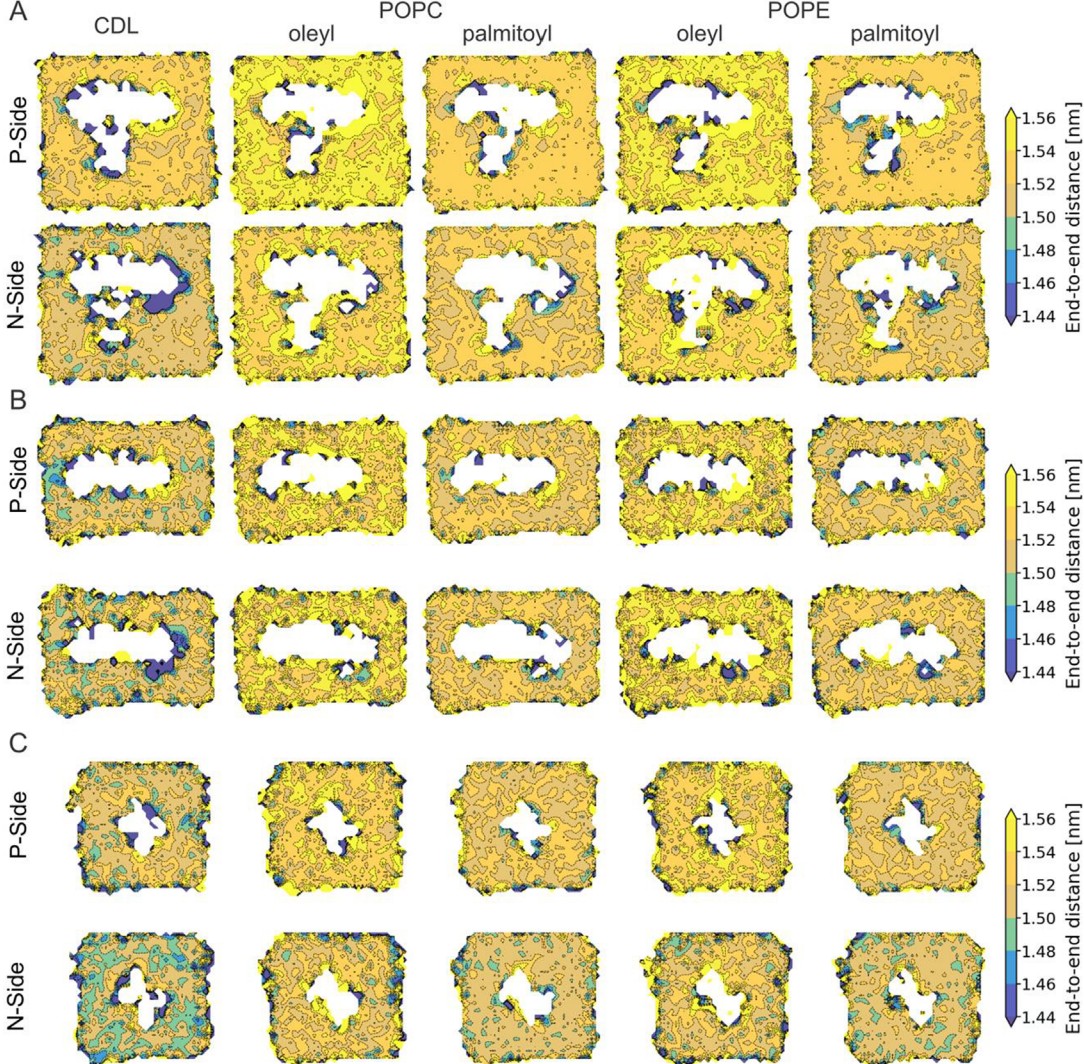

**Appendix 1—figure 6.** Analysis of lipid *end-to-end* distance from atomistic molecular dynamics (aMD) simulations of (**A**) SC, (**B**) CI, and (**C**) CIII$_2$.

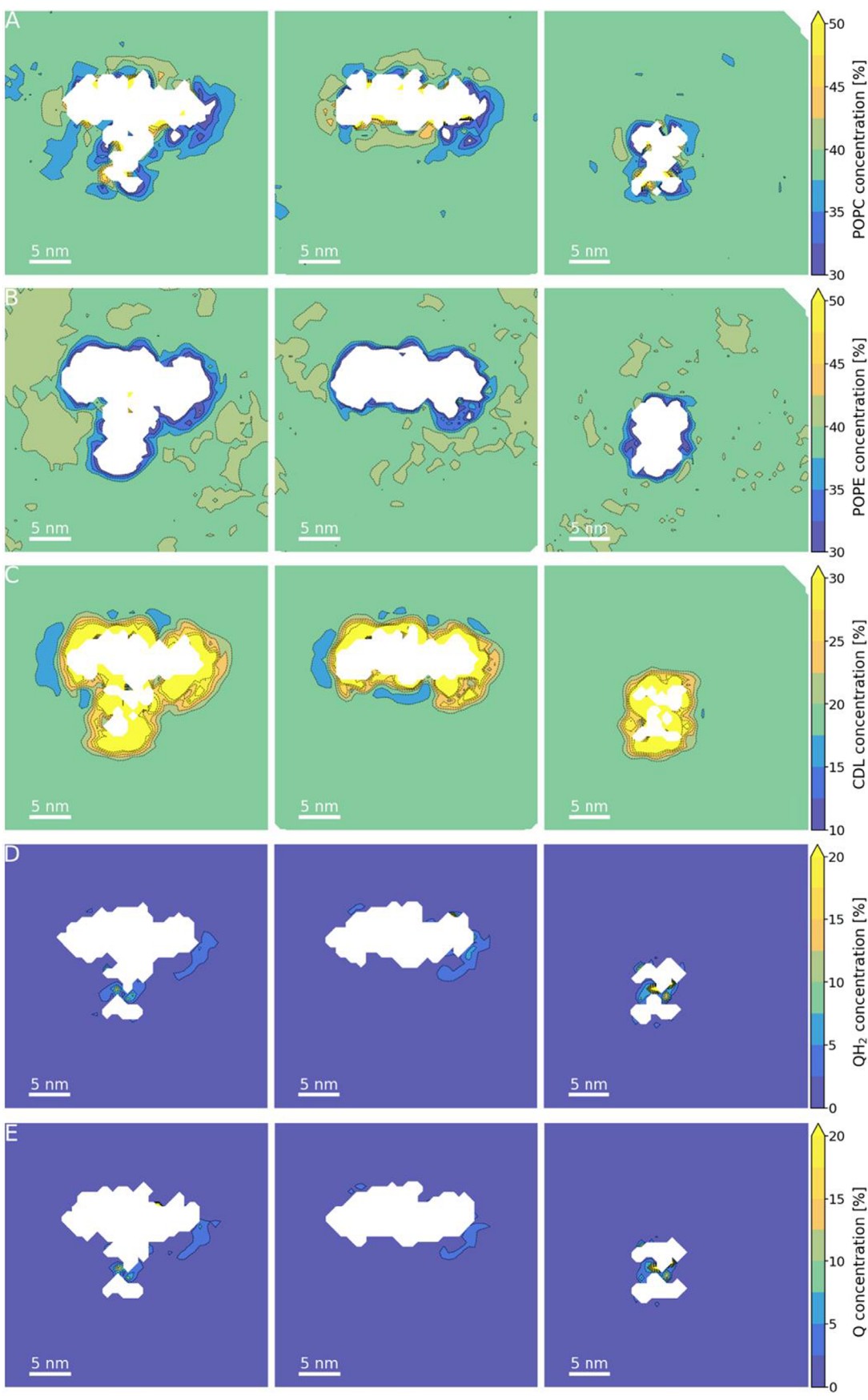

*Appendix 1—figure 7 continued on next page*

*Appendix 1—figure 7 continued*
**Appendix 1—figure 7.** Local lipid concentrations from coarse-grained molecular dynamics (cgMD) simulations. Concentration of (**A**) POPC, (**B**) POPE, (**C**) CDL, (**D**) $QH_2$, and (**E**) Q from cgMD simulations of the $SCI/III_2$ (*left*), CI (*middle*), and CIII (*right*).

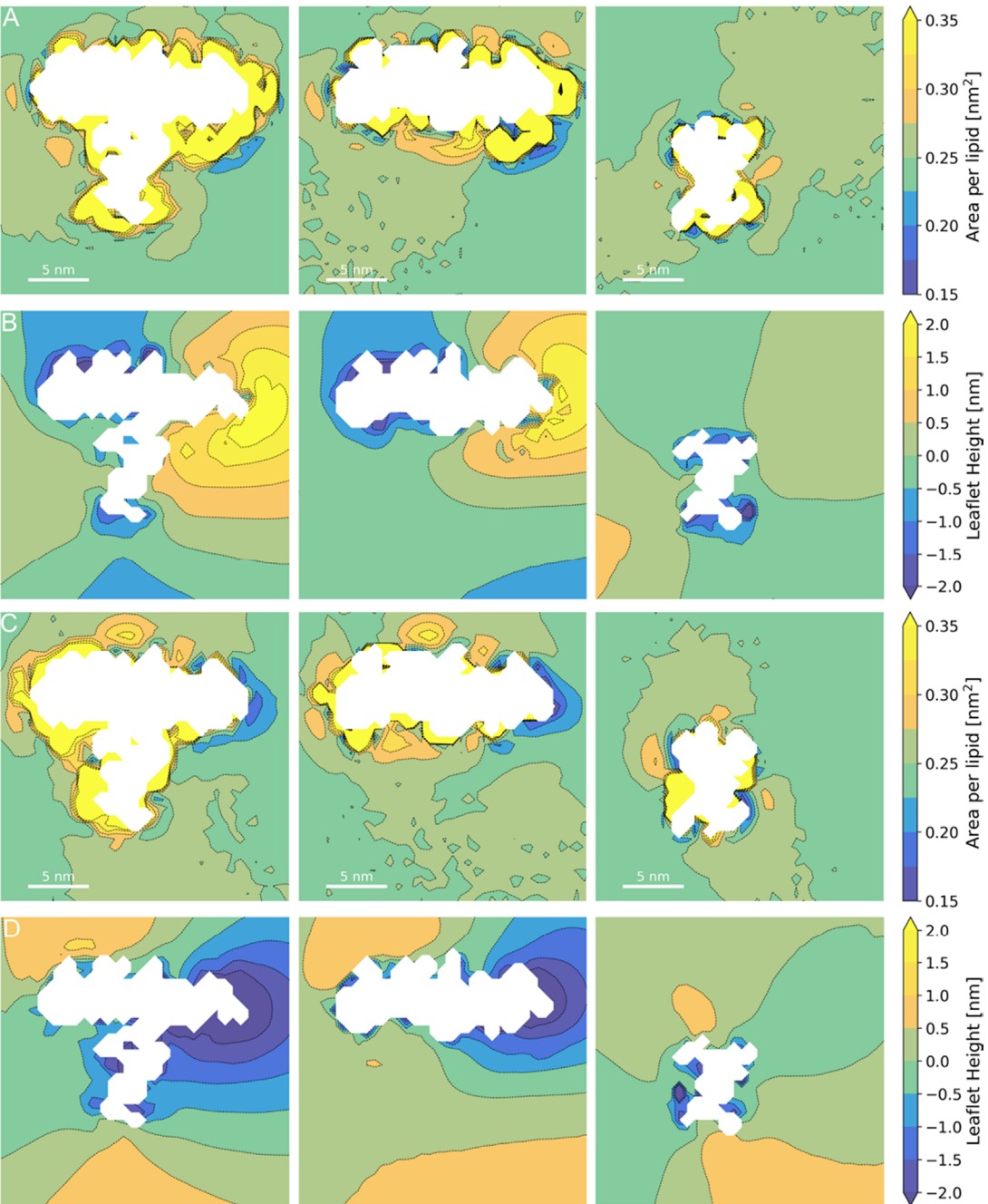

**Appendix 1—figure 8.** Surface area per lipid and local distortion of membrane leaflet. (**A**) The area per lipid on the P-side of the membrane. (**B**) The local leaflet shift in the P-side of the membrane. (**C**) The area per lipid on the N-side of the membrane. (**D**) The local leaflet shift in the N-side of the membrane.

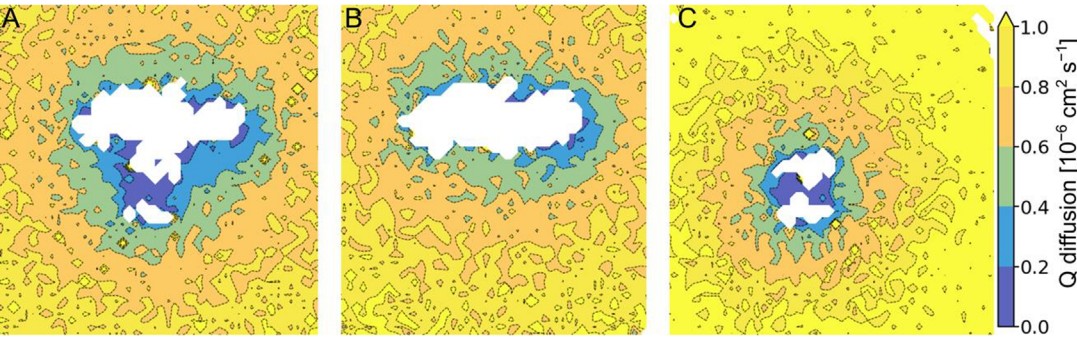

**Appendix 1—figure 9.** Diffusion of Q molecules in coarse-grained molecular dynamics (cgMD) simulations. Map of Q diffusion for (**A**) SCI/III$_2$, (**B**) CI, and (**C**) CIII$_2$.

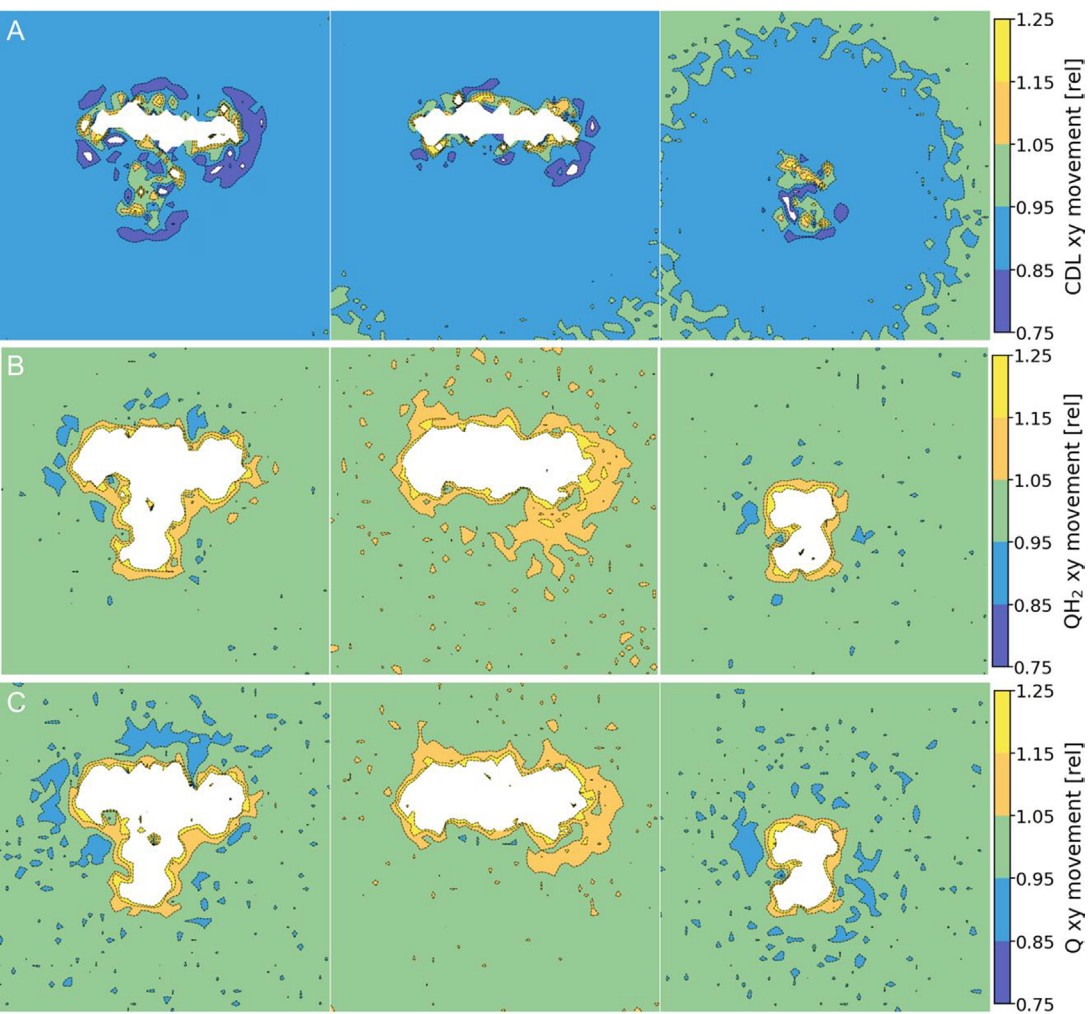

**Appendix 1—figure 10.** Relative movement of lipids in coarse-grained molecular dynamics (cgMD) simulations. Map of average movement of (**A**) CDL and (**B**) QH$_2$ for SCI/III$_2$ (*left*), CI (*middle*), and CIII$_2$ (*right*). The movement is normalized relative to the local movement of POPC and POPE.

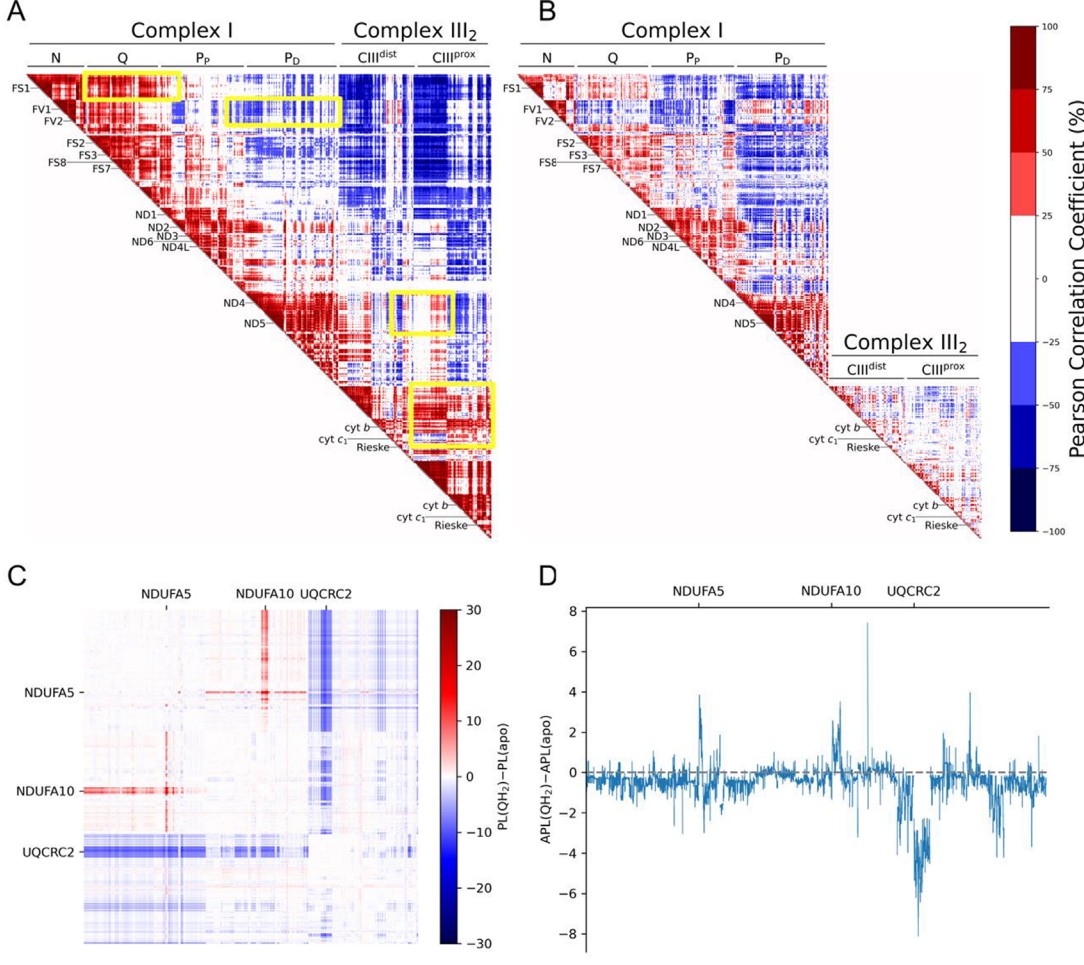

**Appendix 1—figure 11.** Correlated motion within the SC and allosteric network analysis. (**A, B**) The Pearson correlation coefficients from Cα positions during atomistic molecular dynamics (aMD) simulations for (**A**) within the SCI/III$_2$. (**B**) Correlation within the individual CI and CIII$_2$. Core subunits are indicated by labels. Differences between A and B are indicated by yellow boxes. (**C, D**) Allosteric network analysis (see Materials and methods). (**C**) Path length difference of SCI/III$_2$ network for the CI QH$_2$ bound state and CI *apo* state. (**D**) Difference in average path length between the CI QH$_2$ bound state and CI *apo* state.

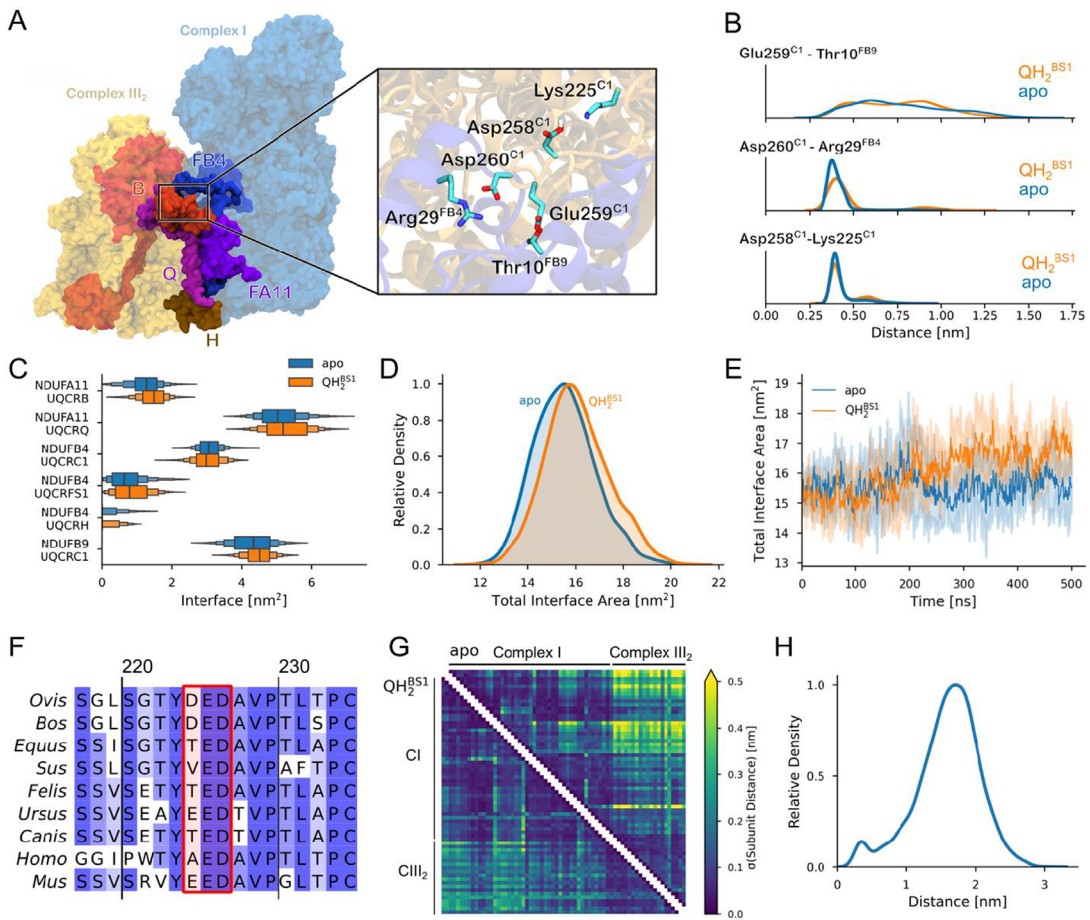

**Appendix 1—figure 12.** Interaction area between CI and CIII$_2$ from atomistic molecular dynamics (aMD) simulations. (**A**) Structure of the SCI/III$_2$ interface. *Inset*: Specific interactions at the CI and CIII$_2$ interface. (**B**) Distribution of the pair distances within the SCI/III$_2$. (**C**) Distribution of interface area between different subunits. (**D**) Distribution of the total interface area for simulations in the *apo* and QH$_2$-bound states of CI. (**E**) Total interface area as a function of simulation time. (**F**) Multiple sequence alignment of UQCRC1, with the carboxylate motif forming the SC contact highlighted in red. (**G**) Standard deviation of the inter-subunit distance matrix for the CI *apo* state (upper triangle) and CI QH$_2$ bound state (lower triangle). (**H**) Distance distribution between C-terminus of NDUFB7 and CIII$_2$. No specific contacts form between these regions during the MD simulations, supporting that substitution of this region resulted in unaltered SCs.

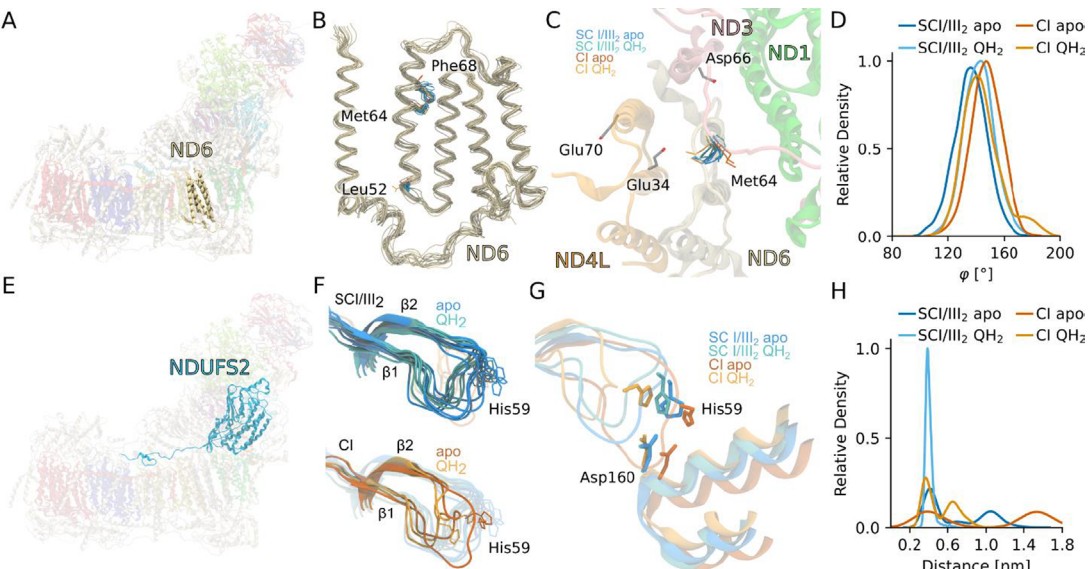

**Appendix 1—figure 13.** Analysis of the TM3$^{ND6}$ dihedral angle in CI. (**A**) Position of the ND6 subunit within CI. Average conformation of ND6 from the last 100 ns – (**B**) the side view and (**C**) the top view of ND6. (**D**) Distribution of $\varphi$ for SCI/III$_2$ and CI in the *apo* and QH$_2$-bound states of CI. (**E**) Position of the NDUFS2 subunit of CI. (**F**) Average conformation (over last 100 ns) of the β1–β2 loop of NDUFS2 from MD simulations for SCI/III$_2$ (*top*) and CI (*bottom*). (**G, H**) Distance distribution between His59 and Asp160 of NDUFS2.

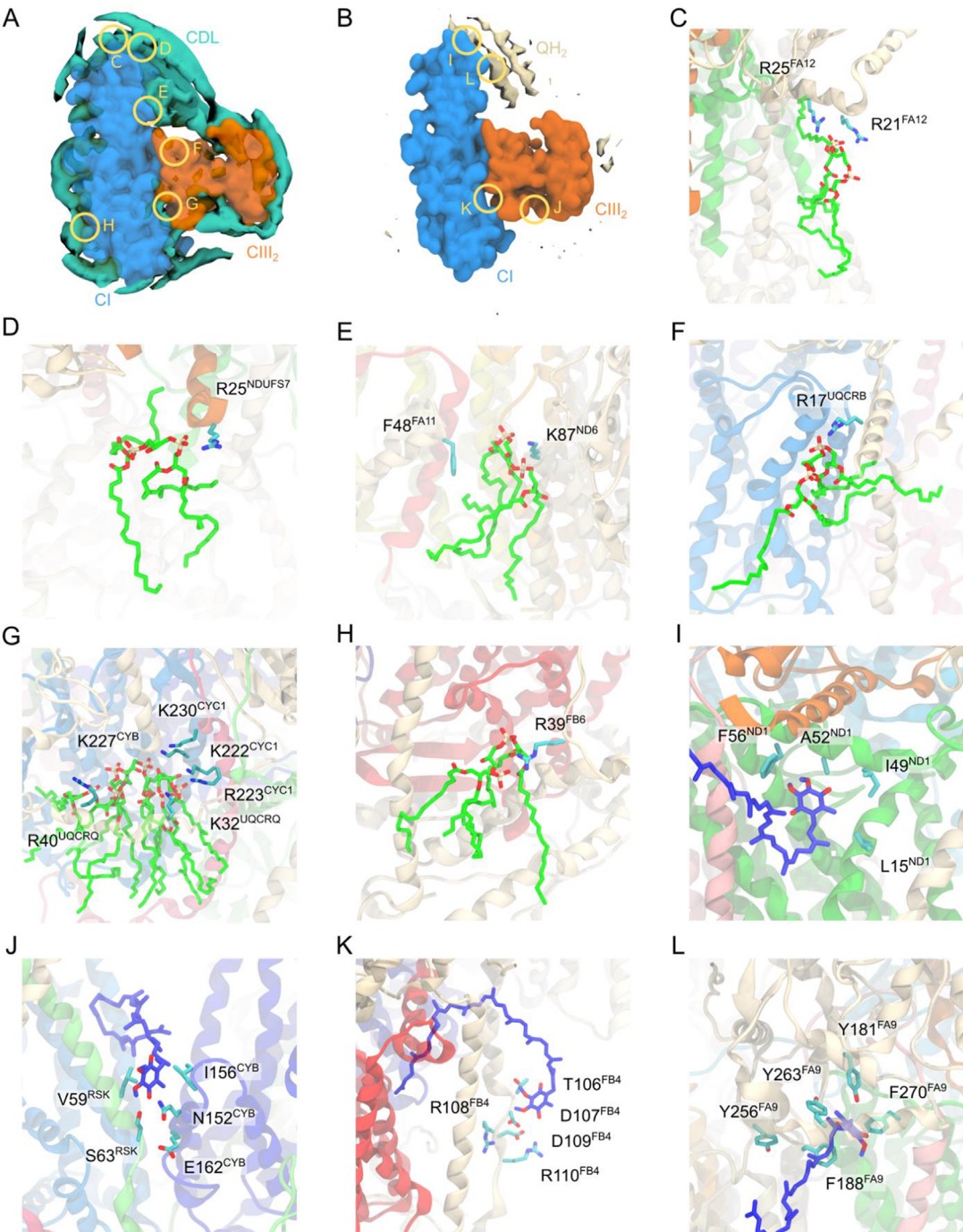

**Appendix 1—figure 14.** Interaction of cardiolipin and quinone/quinol with the SCI/III$_2$. (**A**) Overview of the CDL interaction sites shown in (**C–H**). (**B**) Overview of the Q/QH$_2$ interaction sites shown in (**I–L**).

A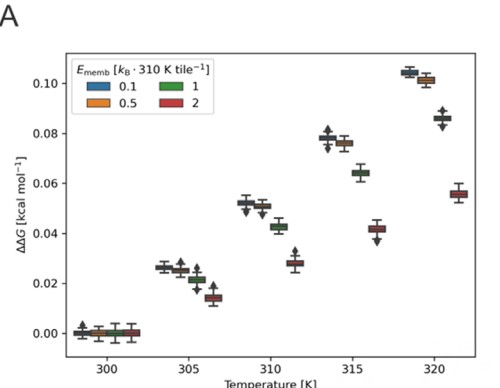 B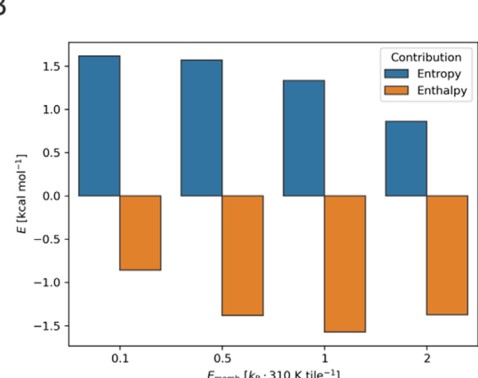

**Appendix 1—figure 15.** Enthalpy–entropy compensations from the lattice model. (**A**) Free energy changes as a function of the introduced membrane strain term. (**B**) Enthalpic and entropic contributions to the free energy as a function of the introduced membrane strain term.

**Appendix 1—table 1.** Estimated membrane deformation energies associated with individual complexes and their supercomplex (SC).

Energies are reported as mean ± standard deviation from bootstrap analysis. The deformation energies are reported for a membrane patch with an area of ca. 1000 nm².

|  | $\Delta G_{\text{curv}}$ (kcal mol⁻¹) | $\Delta G_{\text{thick}}$ (kcal mol⁻¹) |
|---|---|---|
| CI | 123.6 ± 2.8 | 23.9 ± 0.6 |
| CIII$_2$ | 59.6 ± 4.4 | 28.0 ± 1.4 |
| SC | 103.6 ± 1.6 | 49.1 ± 0.9 |
| SC – (CI + CIII$_2$) | –79.2 ± 5.2 | –2.8 ± 2.0 |

**Appendix 1—table 2.** List of atomistic MD and coarse-grained molecular dynamics (cgMD) simulations.

**cgMD**

| Simulation name | Protein model | CI ligand state | CIII proximal ligand state | CIII distal ligand state | Simulation time (µs) |
|---|---|---|---|---|---|
| A1 | SCI/III$_2$ | *apo* | *apo* | *apo* | 2 × 0.5 |
| A2 | | *apo* | Q in Q$_i$, QH$_2$ in Q$_o$ | *apo* | 2 × 0.5 |
| A3 | | *apo* | *apo* | Q in Q$_i$, QH$_2$ in Q$_o$ | 2 × 0.5 |
| A4 | | QH$_2$ | *apo* | *apo* | 2 × 0.5 |
| A5 | | QH$_2$ | Q in Q$_i$, QH$_2$ in Q$_o$ | *apo* | 2 × 0.5 |
| A6 | | QH$_2$ | *apo* | Q in Q$_i$, QH$_2$ in Q$_o$ | 2 × 0.5 |
| A7 | CI | QH$_2$ | *n/a* | *n/a* | 2 × 0.5 |
| A8 | CI | *apo* | *n/a* | *n/a* | 2 × 0.5 |
| A9 | CIII$_2$ | *n/a* | *apo* | *apo* | 2 × 0.5 |
| A10 | | *n/a* | Q in Q$_i$, QH$_2$ in Q$_o$ | *apo* | 2 × 0.5 |
| A11 | | *n/a* | *apo* | Q in Q$_i$, QH$_2$ in Q$_o$ | 2 × 0.5 |
| | | | | Total | 11 µs |

**cgMD**

| Simulation name | Protein model | CI ligand state | CIII proximal ligand state | CIII distal ligand state | Simulation time (µs) |
|---|---|---|---|---|---|

*Appendix 1—table 2 Continued on next page*

*Appendix 1—table 2 Continued*

**cgMD**

| C1 | SCI/III$_2$ | QH$_2$ | Q in Q$_i$, QH$_2$ in Q$_o$ | Q in Q$_i$, QH$_2$ in Q$_o$ | 75 + 50 |
|---|---|---|---|---|---|
| C2 | CI | QH$_2$ | *n/a* | *n/a* | 50 + 40 |
| C3 | CIII$_2$ | *n/a* | Q in Q$_i$, QH$_2$ in Q$_o$ | Q in Q$_i$, QH$_2$ in Q$_o$ | 50 + 50 |
| | | | | Total | 315 µs |

**Appendix 1—table 3.** Simulation details of membrane models.

**aMD**

| Simulation name | System size (Å3) | Lipid composition | Lipid/Q/QH2 molecules | Simulation time (µs) |
|---|---|---|---|---|
| M1 | 78 × 78 × 82 | POPC/POPE (1:1) | 204 | 2 × 0.2 |
| M2 | 78 × 78 × 82 | CDL | 102 | 2 × 0.2 |
| M3 | 187 × 187 × 182 | POPC/POPE/CDL/Q (38:38:19:5) | 1000 | 2 × 0.35 |
| M4 | 187 × 187 × 182 | POPC/POPE/CDL/QH$_2$ (38:38:19:5) | 1000 | 2 × 0.35 |

**cgMD**

| Simulation name | System | Lipid bead distance (nm) | Simulation time (µs) |
|---|---|---|---|
| cgM1/2/3 | SCI/III$_2$ | 0.44/0.50/0.53 | 5/5/5 |
| cgM4/5/6 | CI | 0.44/0.50/0.53 | 5/5/5 |
| cgM7/8/9 | CIII$_2$ | 0.44/0.50/0.53 | 5/5/5 |
| cgM10–13 | Membrane | 0.44/0.47/0.50/0.53 | 5/23/5/5 |

**Appendix 1—table 4.** Non-standard protonation states used in atomistic molecular dynamics (aMD) simulations.

*The protonation states of H59$^{NDUFS2}$ and Y108$^{NDUFS2}$ were modeled in their neutral (His$^0$)/ deprotonated (TyrO$^-$) states with QH$_2$ in CI.

| Subunit | Residues |
|---|---|
| ND1 | E192, E206, H247(ε), H287(ε) |
| ND2 | H25(ε), K46, H48(ε), H112(ε), H186(ε), H232(ε), K263 |
| ND4 | H82(ε), H213(ε/δ), H220(ε), Lys283, H293(ε), H319(ε), H338(ε/δ), H419(ε), H422(ε) |
| ND5 | H27(ε), H56(ε), H109(ε), K119, H230(ε), H248(ε), H323(ε), H348(ε), K392, H484(ε), H509(ε), H605(ε) |
| NDUFS2 | H55(ε), H59(ε/δ)*, Y108*, D104, H150(ε), H157(ε), H190(ε), H200(ε), D292, E343, H348(ε) |
| ND3 | D66, E68, E105 |
| NDUFS7 | D68, E154 |
| NDUFS3 | H19(ε), H53(ε), H145(ε) |
| NDUFV2 | H9(ε), H42(ε), H99(ε) |
| NDUFV1 | H29(ε), D98, H113(ε), H116(ε), H261(ε/δ), H283(ε), H356(ε/δ), H437(ε) |
| NDUFS1 | H43(ε), D232, H255(ε), H293(ε/δ), D324, E347, H401(ε), H421(ε), H437(ε), H494(ε), H549(ε) |
| NDUFS8 | H65(ε/δ), K81, H144(ε/δ) |
| ND6 | E100 |
| ND4L | H52(ε/δ) |
| 18 kDa | H29(ε) |

*Appendix 1—table 4 Continued on next page*

*Appendix 1—table 4 Continued*

| Subunit | Residues |
|---|---|
| 9 kDa | H43($\epsilon$), H44($\epsilon$), H75($\epsilon$) |
| B8 | H21($\epsilon$) |
| B12 | E29 |
| B17 | H67($\epsilon$), H74($\epsilon$), H83($\epsilon$), H89($\epsilon$), H127($\epsilon$/$\delta$) |
| B18 | H3($\epsilon$), H60($\epsilon$), H81($\epsilon$/$\delta$), H84($\epsilon$), Asp87, Glu90, H91($\epsilon$) |
| B22 | H11($\epsilon$), H25($\epsilon$), H32($\epsilon$), H50($\epsilon$), H72($\epsilon$), H75($\epsilon$), H107($\epsilon$) |
| AGGG | H6($\epsilon$), H42 ($\epsilon$/$\delta$), H50($\epsilon$) |
| ASHI | H66($\epsilon$), H78($\epsilon$), H104($\epsilon$), H155($\epsilon$) |
| ESSS | H45($\epsilon$) |
| MNLL | H10($\epsilon$), H13($\epsilon$) |

**Appendix 1—table 5.** Estimation of volume changes from coarse-grained molecular dynamics (cgMD) simulations.

The volume change of the membrane with embedded OXPHOS proteins. The SCI/III$_2$ formation leads to the membrane strain relative to the individual CI and CIII$_2$.

| System | Membrane volume (nm³) | | | $\Delta V$ (nm³) |
|---|---|---|---|---|
| Membrane | 4160 | ± | 1.0 | |
| CI | 3946 | ± | 0.8 | 214 |
| CIII$_2$ | 4093 | ± | 0.7 | 68 |
| SCI/III$_2$ | 3865 | ± | 0.7 | 296 |
| $\Delta V$ | | | | 14 nm³ |

**Appendix 1—table 6.** Estimation of protein copy numbers and effective surface area in the IMM. The data is based on *Schlame, 2021*, see also *Petrache et al., 2000*; *Morgenstern et al., 2021*; *Fedor and Hirst, 2018*. The crowding model, with a square length of 163 nm, describes the flat membrane regions of the IMM, thus excluding the area of ATP synthase and 10 lipids bound to the c-ring, located at the cristae (cf. *Petrache et al., 2000*).

| Component | Copy number | Surface area/molecule (nm²) | Total surface area (nm²) |
|---|---|---|---|
| CI | 13 | 190 | 2470 |
| CII | 15 | 16 | 240 |
| CIII$_2$ | 18 | 110 | 1980 |
| CIV | 64 | 64 | 4096 |
| ATP synthase | 21 | 200 | 4200 |
| ATP/ADP carrier | 160 | 10 | 1600 |
| Lipids | 48,300 | 0.7 | 16,905/leaflet |
| of which Q/QH$_2$ 1% | 483 | | |
| | 291 proteins<br>48,300 membrane | | 14,586 nm² protein<br>16,905 nm² membrane<br>Total: 31,491 nm² |

**Appendix 1—table 7.** Energy terms in the lattice model.

The protein–protein interaction is described by specific interactions term ($E_{specific}$ <0 $k_B T$) and non-specific interactions ($E_{non\text{-}specific}$ >0). The membrane–protein interaction determines the strain energy of the membrane ($E_{strain}$), based on the number of neighboring 'lipid' occupied grids that are in contact with proteins (*Figure 4A*). The interaction between the lipids was indirectly accounted for

by the background energy of the model. The proteins can occupy four unique orientations on a grid ([North, East, South, West]). The table summarizes the unique energies linked to the respective microstates.

| State | CI and CIII$_2$ position | $d$(CI – CIII$_2$) | Energy |
|---|---|---|---|
| 1 | Neighbors with specific interaction | 1 | $E_{specific} + 10\,E_{strain}$ |
| 2 | Neighbors with non-specific interaction | 1 | $E_{non\text{-}specific} + 10\,E_{strain}$ |
| 3 | Diagonal neighbors | $\sqrt{2}$ | $12\,E_{strain}$ |
| 4 | Separated by one lattice position in either cardinal direction | 2 | $13\,E_{strain}$ |
| 5 | Separated by one lattice position in cardinal direction | $\sqrt{5}$ | $14\,E_{strain}$ |
| 6 | Diagonal neighbors separated by one lattice position | $2\sqrt{2}$ | $15\,E_{strain}$ |
| 7 | No interaction | $> 2\sqrt{2}$ | $16\,E_{strain}$ |

**Appendix 1—table 8.** Monte Carlo (MC) simulations of the lattice model.
The conformational landscape was sampled by MC using $10^7$ MC iterations with 100 replicas. Temperature effects were modeled by varying $\beta$, and the effect of different *protein*-to-*lipid* ratios by increasing the grid area. The following simulations were performed (energy units are given in $k_B \times$ 310 K).

| Simulation | Temperature (K) | Grid size, $N$ | $E_{specific}$ | $E_{non\text{-}specific}$ | $E_{strain}$ |
|---|---|---|---|---|---|
| G1–G5 | 300, 305, …, 320 | 4 | −1 | 1 | 0.1 |
| G6–G10 | 300, 305, …, 320 | 4 | −1 | 1 | 0.5 |
| G11–G15 | 300, 305, …, 320 | 4 | −1 | 1 | 1 |
| G16–G20 | 300, 305, …, 320 | 4 | −1 | 1 | 2 |

