## [Editor Report · eLife Assessment]

In this **important** study, the authors conducted extensive atomistic and coarse-grained simulations as well as a lattice Monte Carlo analysis to probe the driving force and functional impact of supercomplex formation in the inner mitochondrial membrane. The study highlighted the major contribution from membrane mechanics to the supercomplex formation and revealed interesting differences in structural and dynamical features of the protein components upon complex formation. Upon revision, the analysis is considered **solid**, although the magnitude of estimated membrane deformation energies seem somewhat large. Overall, the study is thorough, creative and the impact on the field of bioenergetics is expected to be significant.

---

## [Referee Report · Reviewer #1 (Public review)]

This paper by Poverlein et al reports the substantial membrane deformation around the oxidative phosphorylation super complex, proposing that this deformation is a key part of super complex formation. I found the paper interesting and well-written.

* Analysis of the bilayer curvature is challenging on the fine lengthscales they have used and produces unexpectedly large energies (Table 1). Additionally, the authors use the mean curvature (Eq. S5) as input to the (uncited, but it seems clear that this is Helfrich) Helfrich Hamiltonian (Eq. S7). If an errant factor of one half has been included with curvature, this would quarter the curvature energy compared to the real energy, due to the squared curvature. The bending modulus used (ca. 5 kcal/mol) is small on the scale of typically observed biological bending moduli. This suggests the curvature energies are indeed much higher even than the high values reported. Some of this may be due to the spontaneous curvature of the lipids and perhaps the effect of the protein modifying the nearby lipids properties.

* It is unclear how CDL is supporting SC formation if its effect stabilizing the membrane deformation is strong or if it is acting as an electrostatic glue. While this is a weakness for a definite quantification of the effect of CDL on SC formation, the study presents an interesting observation of CDL redistribution and could be an interesting topic for future work.

In summary, the qualitative data presented are interesting (especially the combination of molecular modeling with simpler Monte Carlo modeling aiding broader interpretation of the results). The energies of the membrane deformations are quite large. This might reflect the roles of specific lipids stabilizing those deformations, or the inherent difficulty in characterizing nanometer-scale curvature.

---

## [Referee Report · Reviewer #3 (Public review)]

Summary:

In this contribution, the authors report atomistic, coarse-grained and lattice simulations to analyze the mechanism of supercomplex (SC) formation in mitochondria. The results highlight the importance of membrane deformation as one of the major driving forces for the SC formation, which is not entirely surprising given prior work on membrane protein assembly, but certainly of major mechanistic significance for the specific systems of interest.

Strengths:

The combination of complementary approaches, including an interesting (re)analysis of cryo-EM data, is particularly powerful, and might be applicable to the analysis of related systems. The calculations also revealed that SC formation has interesting impacts on the structural and dynamical (motional correlation) properties of the individual protein components, suggesting further functional relevance of SC formation. In the revision, the authors further clarified and quantified their analysis of membrane responses, leading to further insights into membrane contributions. They have also toned down the decomposition of membrane contributions into enthalpic and entropic contributions, which is difficult to do. Overall, the study is rather thorough, highly creative and the impact on the field is expected to be significant.

Weaknesses:

Upon revision, I believe the weakness identified in previous work has been largely alleviated.

---

## [Author Response]

The following is the authors’ response to the previous reviews

**Reviewer #1 (Public review):**
This paper by Poverlein et al reports the substantial membrane deformation around the oxidative phosphorylation super complex, proposing that this deformation is a key part of super complex formation. I found the paper interesting and well-written.

We thank the Reviewer for finding our work interesting.

Analysis of the bilayer curvature is challenging on the fine lengthscales they have used and produces unexpectedly large energies (Table 1). Additionally, the authors use the mean curvature (Eq. S5) as input to the (uncited, but it seems clear that this is Helfrich) Helfrich Hamiltonian (Eq. S7). If an errant factor of one half has been included with curvature, this would quarter the curvature energy compared to the real energy, due to the squared curvature.

We thank the Reviewer for raising this important issue. We have now clarified in the SI and main manuscript that we employ the Helfrich model. In our initial implementation, we indeed used the mean curvature H, thereby missing a factor of 2. As the Reviewer correctly noted, this resulted in curvature deformation energies that were underestimated by a factor of ~4. We have now corrected for this effect in the revised analysis, and the updated Table 1. Importantly, however, this correction does not alter the general conclusions of our work that supercomplex formation relieves membrane strain and stabilizes the system. We have added an additional paragraph where we discuss the magnitude of the observed bending effects, and compared the previous estimates in literature:

SI:

“The local mean curvature of the membrane midplane was computed using the Helfrich model (4,5) …”

(4) W. Helfrich, Elastic properties of lipid bilayers theory and possible experiments. Zeitschrift für Naturforschung 28c, 693-703 (1973).

(5) F. Campelo et al., Helfrich model of membrane bending: From Gibbs theory of liquid interfaces to membranes as thick anisotropic elastic layers. Advances in Colloid and Interface Science 208, 25-33 (2014).

Main Text:

“which measures the energetic cost of deforming the membrane from a flat geometry (ΔG_curv_) based on the Helfrich model (45, 46). …

Our analysis suggests that both contributions are substantially reduced upon formation of the SC, with the curvature penalty decreasing by 79.2 ± 5.2 kcal mol^-1^ (for a membrane area of ca. 1000 nm^2^) and the thickness penalty by 2.8 ± 2.0 kcal mol^-1^ (Table 1).”

“We note that the magnitude of the estimated bending energies (~10² kcal mol^-1^) (Table 1), while seemingly high at first glance, falls within the range expected for large-scale membrane deformation processes induced by large multi-domain proteins. For example, the Piezo mechanosensitive channel performs roughly 150k_B_T (≈ 90 kcal mol⁻¹) of work to bend the bilayer into its dome-like shape (65). Comparable energies have also been estimated for the nucleation of small membrane pores (66), while vesicle formation typically requires bending energies on the order of 300 kcal mol^-1^, largely independent of vesicle size (67). When normalized by the affected membrane area (~1000 nm^2^), these values correspond to an energy density of approximately 0.1 kcal mol^-1^ nm^-2^, which places our estimates within a biophysically reasonable regime. Notably, cryo-EM structures of several supercomplexes shows that such assemblies can impose significant curvature on the surrounding bilayer (36, 50, 68), supporting the notion that respiratory chain organization is closely coupled to local membrane deformation. Nevertheless, we expect that the absolute deformation energies may be overestimated, as the continuum Helfrich model neglects molecular-level effects such as lipid tilt and local rearrangements, which can partially relax curvature stresses and reduce the effective bending penalty near protein–membrane interfaces (69, 70).”

The bending modulus used (ca. 5 kcal/mol) is small on the scale of typically observed biological bending moduli. This suggests the curvature energies are indeed much higher even than the high values reported. Some of this may be due to the spontaneous curvature of the lipids and perhaps the effect of the protein modifying the nearby lipids properties.

The SI initially included an incorrect value for the bending modulus (20 kJ mol^-1^ instead of 20k_B_T), which has now been corrected. The revised value is consistent with experimentally reported bending moduli from X-ray scattering measurements, although there remains substantial uncertainty in the precise values across different experimental and computational studies.

“The bending deformation energy was computed from the mean curvature field H(x,y), assuming a constant bilayer bending modulus κ (taken as 20k_b_T = 11.85 kcal mol^-1^ (6)):”

(6) S. Brown et al., Comparative analysis of bending moduli in one-component membranes via coarsegrained molecular dynamics simulations. Biophysical Journal 124, 1–13 (2025).

It is unclear how CDL is supporting SC formation if its effect stabilizing the membrane deformation is strong or if it is acting as an electrostatic glue. While this is a weakenss for a definite quantification of the effect of CDL on SC formation, the study presents an interesting observation of CDL redistribution and could be an interesting topic for future work.

We agree with the Reviewer that future studies would be important to investigate the relationship between CDL-induced stabilization of membrane and its electrostatic effects.

In summary, the qualitative data presented are interesting (especially the combination of molecular modeling with simpler Monte Carlo modeling aiding broader interpretation of the results). The energies of the membrane deformations are quite large. This might reflect the roles of specific lipids stabilizing those deformations, or the inherent difficulty in characterizing nanometer-scale curvature.

We thank the Reviewer for appreciating our work and for the help in further improving our findings.

**Reviewer #3 (Public review):**
Summary:In this contribution, the authors report atomistic, coarse-grained and lattice simulations to analyze the mechanism of supercomplex (SC) formation in mitochondria. The results highlight the importance of membrane deformation as one of the major driving forces for the SC formation, which is not entirely surprising given prior work on membrane protein assembly, but certainly of major mechanistic significance for the specific systems of interest.

We thank Reviewer 3 for appreciating the importance of our study.

Strengths:The combination of complementary approaches, including an interesting (re)analysis of cryo-EM data, is particularly powerful, and might be applicable to the analysis of related systems. The calculations also revealed that SC formation has interesting impacts on the structural and dynamical (motional correlation) properties of the individual protein components, suggesting further functional relevance of SC formation. In the revision, the authors further clarified and quantified their analysis of membrane responses, leading to further insights into membrane contributions. They have also toned down the decomposition of membrane contributions into enthalpic and entropic contributions, which is difficult to do. Overall, the study is rather thorough, highly creative and the impact on the field is expected to be significant.Weaknesses:Upon revision, I believe the weakness identified in previous work has been largely alleviated.

We thank the Reviewer for their previous remarks, which allowed us to significantly improve our manuscript.